

# Asymptotes of the nonlinear transfer and wave spectrum

# in the frame of the kinetic equation solution

Vladislav G. Polnikov[1], Fangli Qiao[2, 3] and Yong Teng[2]

[1]A.M. Obukhov Institute of Atmospheric Physics of RAS, Moscow, 119017, Russia
[2]First Institute of Oceanography of SOA, Qingdao, 266061, China
[3]Laboratory for Regional Oceanography and Numerical Modeling, Qingdao National Laboratory for Marine Science and Technology, Qingdao, 266061, China

*Correspondence to*: Vladislav G. Polnikov (polnikov@mail.ru)

**Abstract.** The kinetic equation for a gravity wave spectrum is solved numerically to study the high frequencies asymptotes for the one-dimensional nonlinear energy transfer and the variability of spectrum parameters that accompany the long-term evolution of nonlinear waves. The cases of initial two-dimensional spectra $S(\omega,\theta)$ of modified JONSWAP type with the frequency decay-law $S(\omega) \sim \omega^{-n}$ (for $n$ = 6, 5, 4 and 3.5) and various initial functions of the angular distribution are considered. It is shown that at the first step of the kinetic equation solution, the nonlinear energy transfer asymptote has the
power-like decay-law, $Nl(\omega) \sim \omega^{-p}$, with values $p \leq n-1$, valid in cases when $n \geq 5$, and the difference, $n-p$, changes significantly when $n$ approaches 4. On time scales of evolution greater than several thousands of initial wave periods, in every case, a self-similar spectrum $S_{sf}(\omega,\theta)$ is established with the frequency decay-law of form $S(\omega) \sim \omega^{-4}$. Herein, the asymptote of nonlinear energy transfer becomes negative in value and decreases according to the same law (i.e., $Nl(\omega) \sim -\omega^{-4}$). The peak frequency of the spectrum, $\omega_p(t)$, migrates in time $t$ to the low-frequency region such that the angular and
frequency characteristics of the two-dimensional spectrum $S_{sf}(\omega,\theta)$ remain constant. However, these characteristics depend on the degree of angular anisotropy of the initial spectrum. The solutions obtained are interpreted, and their connection with the analytical solutions of the kinetic equation by Zakharov and co-authors for gravity waves in water is discussed.

## 1 Introduction

In the study of random field features for the nonlinear gravity waves in water, the four-wave kinetic equation (KE)

$$\frac{\partial N(\mathbf{k}_0)}{\partial t} = I_{NL}(N) \equiv$$

$$\equiv 4\pi \int M^2(\mathbf{k}_0,\mathbf{k}_1,\mathbf{k}_2,\mathbf{k}_3)\{N(\mathbf{k}_2)N(\mathbf{k}_3)[N(\mathbf{k}_0)+N(\mathbf{k}_1)]-N(\mathbf{k}_0)N(\mathbf{k}_1)[N(\mathbf{k}_2)+N(\mathbf{k}_3)]\}\delta_{0+1-2-3}d\mathbf{k}_1 d\mathbf{k}_2 d\mathbf{k}_3$$
(1)

plays a very important role. In (1), $I_{NL}(N)$ is the kinetic integral (KI), $N(\mathbf{k}_i)$ is the two-dimensional wave-action spectrum, written in the wave vector **k**-space ($i$=0,1,2,3), $M^2(\mathbf{k}_0,\mathbf{k}_1,\mathbf{k}_2,\mathbf{k}_3)$ is the second power of the matrix elements



corresponding to the four-wave nonlinear interactions, $\delta_{0+1-2-3} \equiv \delta(\omega_0 + \omega_1 - \omega_2 - \omega_3)\delta(\mathbf{k}_0 + \mathbf{k}_1 - \mathbf{k}_2 - \mathbf{k}_3)$ is the Dirac delta-function responsible for the resonant feature of the four-wave interactions, and $\omega_i = \omega(\mathbf{k}_i)$ is the radian frequency of the wave component with wave vector $\mathbf{k}_i$. The relationship between wave vector $\mathbf{k} = (k_x, k_y) = (k, \theta)$ and wave frequency $\omega$ is given by the dispersion relation. For gravity waves in deep water, considered here, it is; $\omega(\mathbf{k}) \equiv \omega(k) = (gk)^{1/2}$, where $g$ is

the gravity acceleration. Later we use KE (1) in the form written for the wave-energy spectrum, $S(\omega, \theta)$, given in the frequency-angular space, $(\omega, \theta)$, and related linearly to $N(\mathbf{k})$ (e.g., Badulin et al., 2005)

$$S(\omega, \theta) = \omega N(\omega, \theta) = 2(\omega^4 / g^2)N(\mathbf{k}) \qquad . \qquad (2)$$

The expressions for matrix elements $M_{0,1,2,3}$ are well known (e.g., Hasselmann, 1962; Webb, 1968; Zakharov, 1968; see also Badulin et al., 2005), but we use the elements from Crawford et al. (1980). In any representation, formulas for elements

$M_{0,1,2,3}$ are very cumbersome, so they are not given here. It is important to note only that the KI formally preserves the total wave energy,

$$E = \iint S(\omega, \theta)d\omega d\theta, \qquad (3)$$

the total wave action,

$$N = \iint (S(\omega, \theta) / \omega)d\omega d\theta, \qquad (4)$$

and the total wave momentum,

$$\mathbf{M} = \iint (\mathbf{k}S(\omega, \theta) / \omega)d\omega d\theta, \qquad (5)$$

under the condition of uniform convergence of the KI (e.g., Hasselmann 1963; Zakharov et al. 1992).

Equation (1) was first derived by Hasselmann (1962) in the potential approximation, starting from the Euler equations for waves in water. Soon, Zakharov (1966) rederived KE (1) using the technique of the Hamiltonian formalism (see also,

Zakharov et al. 1992; Badulin et al., 2005). Since late 60s, the KE became a separate subject of studying (e.g., Zakharov and Filonenko, 1966; Zakharov and Zaslavskii, 1982; Webb, 1978; Masuda, 1980; Hasselmann and Hasselmann, 1981; Hasselmann et al., 1985; Polnikov, 1989, 1990, 1996, 2007; Resio and Perrie, 1991; Pushkarev et al., 2003; Badulin et al., 2005; Young and Van Vledder, 2003; Van Vledder, 2006; Badulin and Zakharov (2017); Geodjaev and Zakharov 2017; Zakhariov, 2017, and references therein). Among them, analytical results found by Zakharov and co-authors (see below)

play the crucial role in these investigations.

First, for the case of an angular isotropic spectrum spread throughout the infinite frequency band ($0 < \omega < \infty$), Zakharov and Filonenko (1966) have found the analytic solution of KE (1) of form $S_Z(\omega) \propto \omega^{-4}$, which puts the KI identically to zero (i.e., $I_{NL}[S_Z(\omega)] \equiv 0$). Later, Zakharov and Zaslavskii (1982) found that KE (1) has the second analogous solution of



form $S_{ZZ}(\omega) \propto \omega^{-11/3}$. They have proposed to interpret these solutions as the Kolmogorov-type spectra of the constant energy-flux, $P_E$, directed upward in frequencies, and constant wave-action flux, $P_N$, downward in frequencies, respectively. In this interpretation, the mentioned spectra acquire the proper Kolmogorov-type representation of the form (Zakharov and Zaslavskii, 1982; see also Pushkarev et al., 2003; Badulin et al., 2005)

$$S_Z(\omega) = c_1 P_E^{1/3} g^{4/3} \omega^{-4},$$  (6)

and

$$S_{ZZ}(\omega) = c_2 P_N^{1/3} g^{4/3} \omega^{-11/3},$$  (7)

where $c_1$ and $c_2$ are the dimensionless Kolmogorov's constants. The proper sources and sinks of these fluxes, obligatory in the Kolmogorov's theory (Monin. and Yaglom, 1971), were assumed to be located at the zero and infinity frequency-points, in dependence on the flux respectively (Zakharov and Zaslavskii, 1982). It should be noted, however, that this interpretation was a hypothesis, at that time, because it was based only on dimensional considerations and similarity to the Kolmogorov's theory, but does not follow from the Euler equations without external forces, having no sources and sinks. Later, in a large series of papers by Zakharov and co-authors (see below), the mentioned analytical results were sophisticated in different directions: generalisation for a case of anisotropic spectra in (Zakharov and Pushkarev, 1999), basing on paper by Katz and Kontorovich (1974); description of the spatial and temporal asymptotes of the self-similar spectra of form (6) (Pushkarev et al., 2003; Badulin et al., 2005); and searching for convergence conditions for the KI (Geodjaev and Zakharov 2017; Zakharov 2017). These analytical results are represented in details in the cited papers and in numerous references therein.

In parallel, a study of KE was carried out numerically, beginning with the works by Webb (1978), Masuda (1980), and Hasselmann and Hasselmann (1981), devoted to the development of methods for calculation of the KI. Finally, only two methods have survived: by Webb (1978) and Masuda (1980). The first of them was later elaborated by Resio and Perrie (1991) and Van Vledder (2006). It is used by these authors and the Zakharov's group. The second one elaborated by Polnikov (1989), Lavrenov (2003), Komatsu and Masuda (1996), Gagnaire-Renou, et al, (2010), who use it. Evaluation of these methods was done in (Benoit, 2005). Here it should be noted that the most important part of the KI-calculation algorithm is the method of estimating the contribution of integrable singularities in the integrand (the contribution of 'singular points'), resulting from integration of the frequency delta-function (Masuda 1980; Polnikov 1989). The different approaches to this point represent the main differences in the algorithms mentioned (e.g., Masuda, 1980; Van Vledder, 2006; Polnikov, 1989; Lavrenov, 2003). Herewith, Polnikov (1989) have showed numerically that the contribution of the singular points determines crucially the conservation balance for the integral values $E, N$, **M** (Eqs, 3-5), and the accuracy of the KI-estimate as a whole.

For the first time, a detailed numerical study of the KI properties was carried out by Masuda (1980). It was expanded by Polnikov (1989) via introducing classification for the two-dimensional spectral shapes, $S(\omega,\theta)$. After that, Polnikov (1990) has performed a numerical solution of KE (1), where it was shown the fact of establishing the self-similar shape of wave





spectrum, $S_{sf}(\omega,\theta)$, in the course of long-term evolution of nonlinear waves. This result was later confirmed in a series of papers by the Zakharov's group (see references in Pushkarev et al., 2003; Badulin et al., 2005) and others (e.g., Lavrenov, 2004; Gagnaire-Renou and Benoit, 2007).

In addition to Polnikov (1990), Pushkarev et al. (2003) and Badulin et al. (2005) have established that the tail of the self-similar spectrum $S_{sf}(\omega)$ falls according to the law $S_{sf}(\omega,\theta) \sim \omega^{-4}$, and the shape of spectral peak for $S_{sf}(\omega,\theta)$ is close to one for the JONSWAP spectrum with $\gamma = 3.3$ (Badulin et al., 2005). They first noted that the only condition of conservation for total wave action $N$ takes place, during wave spectrum evolution according to KE (1), and this evolution is accompanied by a leak of the total wave energy $E$ and appearance of a certain energy flux $P_E$ forwarded to the higher frequencies. This result was treated as the formation of the Kolmogorov-like spectrum of form (6) in the frame of solving KE (1).

The mentioned effects represent the most interesting points for further investigations, taking in mind that all three integral values: $E$, $N$, and **M,** should be formally preserved simultaneously, as far as KE (1) is derived from the conservative Euler equations without external forces (Hasselmann, 1963; Zakharov et al, 1992). The present work is devoted to clarification this point in more details. In addition to this, other essential features of the spectral peak shape and the process of forming two-dimensional self-similar spectrum $S_{sf}(\omega,\theta)$ are still not sufficiently described, i.e., the high-frequency asymptotes of the non-

linear energy transfer at short- and long-time evolution, integral parameters of the spectral peak and angular distribution for $S_{sf}(\omega,\theta)$. The mentioned points will be elaborated in the present paper also.

Besides of studying the shape of self-similar spectrum $S_{sf}(\omega,\theta)$, a lot of papers are devoted to a searching for establishing the Kolmogorov-spectra of forms (6) and (7). First, under advice of Zakharov, Polnikov (1994) has showed numerically that these spectra are actually formed as the result of numerical solution for the extended KE having the form

$$\partial S(\omega,\theta)/\partial t = I_{NL}[S(\omega,\theta)] + In(\omega,\theta) - Dis(\omega,\theta). \qquad (8)$$

In (8), $In(\omega,\theta)$ and $Dis(\omega,\theta)$ are the source and sink functions, respectively, which are separated in the frequency space, to ensure the presence of the inertial interval, as the main condition for applicability of the Kolmogorov's theory (Monin and Yaglom, 1971). Despite of technical limitations in performing calculations, these results (Polnikov, 1994) gave grounds for the validity of the hypotheses by Zakharov and Zaslasvskii (1982). Similar, and more detailed results were later obtained in

numerous studies (e.g., Pushkarev et al., 2003; Lavrenov, 2004; Badulin et al., 2005; Badulin and Zakharov, 2017; and references therein) in which other algorithms and thicker numerical grids were used to compute the KI. They confirmed the fact of establishing spectra of forms (6) and (7), depending on the configuration of the source and sink locations on the calculating frequency band. In addition, Pushkarev et al. (2003) and Badulin et al. (2005) have analysed the long-term asymptote of the peak-frequency downshift, $\omega_p(t)$, estimated the Kolmogorov's constant values, and determined the rate of

wave-energy leakage $E(t)$ in the case of preservation of total wave action value $N$. These results were applied in the recent paper by Badulin and Zakharov (2017), devoted to a modelling numerous details of the spectral shape for self-similar spectrum of swell, including the well-known effect of the swell angular bimodality at high frequencies.



Nevertheless, studying the features of the KE of form (1) requires its continuation. Namely, asymptotes of the nonlinear energy transfer, $Nl(\omega,\theta) \equiv I_{NL}\big[S(\omega,\theta)\big]$, at high frequencies, both at the initial stage and on long-term scales of the spectrum evolution, is not yet described. The quantitative characteristics of the two-dimensional self-similar spectrum shape, $S_{sf}(\omega,\theta)$, as well as their dependence on the initial conditions, are not sufficiently specified (despite of the recent paper by

Badulin and Zakharov, 2017). There is also a need to analyze the impact of the conservation condition for total wave energy $E$ (besides of conservation of wave action $N$) on the shape of the self-similar spectrum. This task could be especially realized as a theoretical scenario for the numerical solution of KE (1) without sources and sinks. The comparison of these two cases of the numerical solution scenario for KE is very interesting for understanding the nature of establishing self-similar spectrum $S_{sf}(\omega,\theta)$ mentioned above.

This paper is devoted to studying these details of numerical solution for KE (1). The methods of research and computations are presented in Sect. 2. Calculation results and analysis are given in Sect. 3. Section 4 is devoted to the interpretation of the results, and Sect. 5 contains conclusions. Appendix (A) contains the analytical derivations of asymptotes for the peak frequency downshifting, $\omega_p(t)$, accompanying the self-similar spectrum evolution.

## 2 Methods of research

### 2.1 Scenarios of the KE solution

First of all, let us dwell shortly on the point of the laws of conservation in the frame of KE (1). From the very beginning, it was proved that the Euler equations without external forces are the conservative system (Zakharov, 1968), i.e. the energy of the system does not change in time. Thus, the KE, derived from the mentioned equations, should also preserve wave energy,

at least. Eventually, it was proved that KE (1) formally preserve three integral wave values: energy $E$, action N, and momentum **M**, given by Eqs. (3-5), as was already mentioned. Though, a proper special testing of the known and actively used algorithms for the KI-calculations is not presented in literature in all details, being, usually, the intrinsic matter of the authors (e.g., Masuda, 1980; Polnikov, 1989; Resio and Perrie,1991; Van Vledder, 2006).

Herewith, it is important to note that all the mentioned algorithms provide establishing the self-similar spectrum, $S_{sf}(\omega,\theta)$

(see numerous references above), what could contradicts to the preserving three integral values: values: $E$, $N$, and **M**, as was many times noted (e.g., Pushkarev and Zakharov, 2000; Pushkarev et al, 2003; Badulin et al, 2005; Geodjaev and Zakharov, 2017; Zakharov, 2017). Indeed, let us introduce representation for the one-dimensional self-similar spectrum, $S_{sf}(\omega)$, in the form (we consider the 1D case, for simplicity)

$$S_{sf}(\omega,\omega_p) = S_p F_{sf}(\omega/\omega_p) \qquad , \qquad\qquad\qquad (9)$$

where $S_p \equiv S(\omega_p)$ and $\omega_p(t)$ is the peak frequency depending on time of evolution, whilst $F_{sf}(\omega/\omega_p)$ is the non-dimensional (hereafter, n.d.) self-similar spectral shape constant in time. If spectral shape (9) is self-similar, then the 1D nonlinear





transfer, $Nl(\omega)$, should also be self-similar, as it is the integral from self-similar functions. Thus, one may write

$$Nl_{sf}(\omega, \omega_p) = c \cdot g^{-4} S_p^3 \omega_p^{11} \cdot F_{sfN}(\omega / \omega_p) = c \cdot R_{Nl} \cdot F_{sfN}(\omega / \omega_p) ,$$  (10)

where $c = \pi/16$ is the theoretical coefficient, $R_{Nl} = g^{-4} S_p^3 \omega_p^{11}$ is the dimension of non-linear energy transfer $Nl(\omega)$. (Polnikov, 1989), and $F_{sfN}(\omega / \omega_p)$ is the n.d. self-similar spectral shape of $Nl_{sf}(\omega)$. Then, conditions (3-4) lead to the system of the

following equations (here consider only two values $E$ and $N$, for simplicity)

$$\int F_{sfN}(\sigma) d\sigma = 0, \qquad \int [F_{sfN}(\sigma) / \sigma] d\sigma = 0 , \qquad .$$  (11)

where $\sigma = \omega / \omega_p$ is the n.d. frequency. System (11) gives restrictions for the shape of $F_{sfN}(\sigma)$ (and, implicitly, to $F_{sf}(\sigma)$). The question under consideration is: does a solution of system (11) exist? As function $F_{sfN}(\sigma)$ is prescribed (it is defined by the KI) and has one positive lobe in the range σ ≤ 1 and one negative lobe for σ >1 (as well known, e.g., Pushkarev et al.,

2003; Badulin et al., 2005; and evidently shown below), it follows that system (11) is inconsistent, and simultaneous preservation of two values, $E$ and $N$, is impossible. That is the paradox of the KE, which needs its own explanation. At some extent it is done in (Zakharov, 2017) via the bad convergence of the KI for slow-falling spectra, though it needs more details.

Many years ego, on the basis of specific numerical simulations with KE, it was pointed out that the wave energy should leak to the higher frequencies due to the four-wave nonlinear interactions (Pushkarev and Zakharov, 2000). Since that time

this idea is applied in any case of considering KE (1) by the Zkharov's group (e.g. Pushkarev et al., 2003; Badulin et al., 2005; Geodjaev and Zakharov, 2017; Zakharov, 2017), though, in our mind, there is no convincing proof of it. Usually, these authors say about conservation of $N$ and leakage of $E$ and **M**, do not dwelling on physical reasons of zero-balance mismatch for them in the frame of KE solution. As we have no convincing (say, categorical) prohibition for saving $E$ during the solving of KE (1), in our mind, two scenarios of preservation for both of these quantities may be chosen, at least, as an

interesting theoretical alternative. It occurs that this way gives another paradox result (see below).

**2.2 Numerical details**

In the numerical study of the KI-properties and features of the KE-solution, the choice of a certain frequency-angular numerical grid *(ω,θ)* provides (at a certain extent) the accuracy of calculations (Polnikov, 1999; van Vledder, 2006). In our

study, to examine the asymptote of the non-linear transfer of energy (NLT), the extended frequency grid and enhanced angular resolution are chosen. They are given by the ratios:

0.64 ≤ ω ≤ 51.7 rad/s                    and                    -180° ≤ θ ≤ 180°;                    (12a)

with

$\omega_i = \omega_1 q^{i-1}$        and $\omega_1 = 0.64$ rad/s, $q = 1.05$, $1 \le i \le I = 90$, and $\Delta\theta = 5°$.                    (12b)



The initial high-frequency asymptote of NLT is determined from the results of calculating KI at the first time-step of the KE solution. To study the asymptote of NLT and spectral shape resulting from numerical solutions of KE on long-term scales for spectrum evolution, we have used the truncated frequency grid and lower angular resolution:

$$0.64 \leq \omega \leq 7 \text{ rad/s with } I = 50, \Delta\theta = 10^0, \tag{13}$$

and the previous values for $\omega_1$ and $q$. This was done for the reasons of arising technical difficulties during numerous and long-term calculations on grid (12a,b) (high time consuming and appearing numerical instability).

The initial wave spectrum was given in the slightly modified form for the well-known JONSWAP spectrum (J-spectrum)

$$S_J(\omega,\theta,n,\gamma) = S_{PM}(\omega,n)\gamma^{\left\{(\omega/\omega_p - 1)^2/2\Delta^2\right\}}\Psi(\omega,\theta) \tag{14}$$

where $\Delta = 0.07$ to $0.09$ is the n.d. peak-width parameter of the J-spectrum, $\gamma = 1$ to $7$ is the n.d. 'peakness' parameter, $\Psi(\omega,\theta)$ is the n.d. frequency-angular form, and

$$S_{PM}(\omega,n) = 0.01g^2(\omega^{-n}/\omega_p^{5-n})exp\left[-(n/4)(\omega_p/\omega)^4\right] \tag{15}$$

is the Pearson-Moskowitz spectrum, modified to the case of an arbitrary degree of the spectrum-tail decay, $n$ (following to Polnikov 1989). The initial peak frequency, $\omega_p(0)$, was assumed to be 2 rad/s, whilst the initial form of function $\Psi(\omega,\theta)$ was assumed to be independent of the frequency and given in the form

$$\Psi(\omega,\theta) \equiv \Psi(\theta) = cos^a(\theta/m) . \tag{16}$$

Variation of the parameters $a$ and $m$ ensure the assignment of any degree of angular anisotropy of the spectrum used for calculation of the KI at the initial stage of spectrum evolution.

The algorithm for calculating the KI was used according to work by Polnikov (1989), where it was shown a high smoothness and numerical stability of the proper NLT-estimates. Comparison of the KI-estimates on grids (12a,b) and (13) has showed that the one-step KI-calculations have relative deviations within 3% to 5% (for peak values of the NLT), what gives the upper limits for the errors of calculations.

To check the conservation balance $Q$, we have used the ratio

$$Q_{E,N} = \left(\int\limits_{Nl>0} Nl_{E,N}(\omega,\theta)d\omega d\theta + \int\limits_{Nl<0} Nl_{E,N}(\omega,\theta)d\omega d\theta\right)/\int\limits_{Nl>0} Nl_{E,N}(\omega,\theta)d\omega d\theta , \tag{17}$$

where the sub-indexes $E$ and $N$ mean the cases of considering NLT for the energy spectrum or wave action one, respectively. Our experience shows that for the fast falling spectra (faster $\omega^{-5}$), balance $Q$ has the value about of 1-2% for both values: $E$ and $N$ (**M** was out of our considerations), For the slower falling spectra, the balance for $N$ retains the same, whilst for $E$ it becomes negative and has the order of 5-10% (depending on the shape of spectrum and stage of evolution, see below). The more detailed consideration of this point is out of this paper. It needs a special research, as mentioned already. Nevertheless, the presented estimates are quite acceptable for our purposes to calculate the wave evolution separately: with the exact preservation of $E$, and the same for $N$.


In the KE-solution, the exact balance, $Q_E = 0$ or $Q_N = 0$, is made by correcting the whole negative part of the relevant NLT , $Nl_{E,N}(\omega,\theta)$, with a proper coefficient, at each time-step. It is done for the "purity" of the scenarios mentioned. This correction does not influence on the high frequency asymptotes of NLT, absolutely. Herewith, it saves the physics of spectrum evolution, because the mentioned corrections are weaker than one-step changes of the spectra in the course of the

KE-solution.

The behaviour of the high-frequency asymptote for the one-dimensional NLT, $NL(\omega)$, was determined by the least-squares estimation of the decay parameter $p$ in the power-law dependence for $NL(\omega)$ of the form

$$\int I_{NL}\{S(\omega,\theta,t)\}d\theta \equiv Nl(\omega,t) = const(t) \cdot \omega^{-p}. \tag{18}$$

The estimation of $p$ was executed in frequency band $5\,\omega_p < \omega < (15\text{-}20)\omega_p$ for numerical grid (12). The quantitative

characteristics of the two-dimensional spectrum-shape were determined in the integral representation form by introducing the spectral frequency-width parameter, $B$, and the angular-narrowness function of spectrum, $A(\omega)$ ( Polnikov, 1989, 1990):

$$B = E / \omega_p S_p \qquad \text{and} \quad A(\omega) = S(\omega,\theta_p)/S(\omega). \tag{19}$$

Here, $S_p = S(\omega_p)$ is the value of the one-dimensional spectrum at the peak frequency, and $\theta_p$ is the direction of propagation of the peak-component for a two-dimensional wave spectrum. Hereafter, $\theta_p = 0$.

The numerical solution of KE (1) was performed according to the algorithm by Polnikov (1990), which includes an explicit numerical scheme of the first order of accuracy, linear interpolation of the spectrum from the nodes of the computational grid to the resonance points of four-wave quadruplets, and the choice of the time-step, $\Delta t$, according to the condition

$$\Delta t = 0.03 / min(abs[I_{NL}\{S(\omega,\theta,t)\}/S(\omega,\theta,t)]). \tag{20}$$

Ratio (20) indicates restriction of the spectrum-intensity change to 3% or less of the current value, $S(\omega,\theta,t)$, at each time-step. Although this choice of $\Delta t$ led to a significant increase of computing time, it was adopted, because it ensures smoothness and numerical stability of the solution, resulting in good final results.

In this paper, all calculations were performed in the dimensional units. First, the scenario of preservation for $E$ was realized, in addition to considerations by the Zakhariv's group (see references above), assuming the wave system is the

conservative one. Then, some KE-solutions were done with the scenario of preservation for $N$, to demonstrate the difference between them. Below, these scenarios are named as the algorithms versions for the KE-solution. Analysis of the spectral forms of numerical solutions for KE (1) was carried out at evolution times exceeding $10^5/\omega_p(0)$.




## 3 Calculation results and analysis

### 3.1 Asymptotes of the NLT

The one-step calculations of KI on grid (12) were performed for several spectral forms, the representative set of which is given in Table 1, where the estimates of parameter $p$ for the power asymptote of the NLT of form (18) are also shown. The

5 typical forms of the calculated one-dimensional functions $NL(\omega)$ and their asymptotes are shown in Figs 1a,b.

Table 1. Asymptotes of the NLT at the first time step of KE solution.

| Run | Initial spectrum shape | | | Asymptote |
|---|---|---|---|---|
| No | $n$ | $\gamma$ | $\Psi(\theta)$ | $p$ |
| 1 | 6 | 3.3 | const | 4.9 |
| 2 | 6 | 3.3 | $\cos^2(\theta)$ | 5.3 |
| 3 | 6 | 1.0 | const | 5.2 |
| 4 | 5 | 3.3 | const | 3.3 |
| 5 | 5 | 3.3 | $\cos^2(\theta)$ | 3.8 |
| 6 | 5 | 1.0 | const | 3.8 |
| 7 | 4 | 3.3 | const | 0.85 |

Note. The powers of asymptotes are found in the frequency band $5 < \omega/\omega_p < 15$.

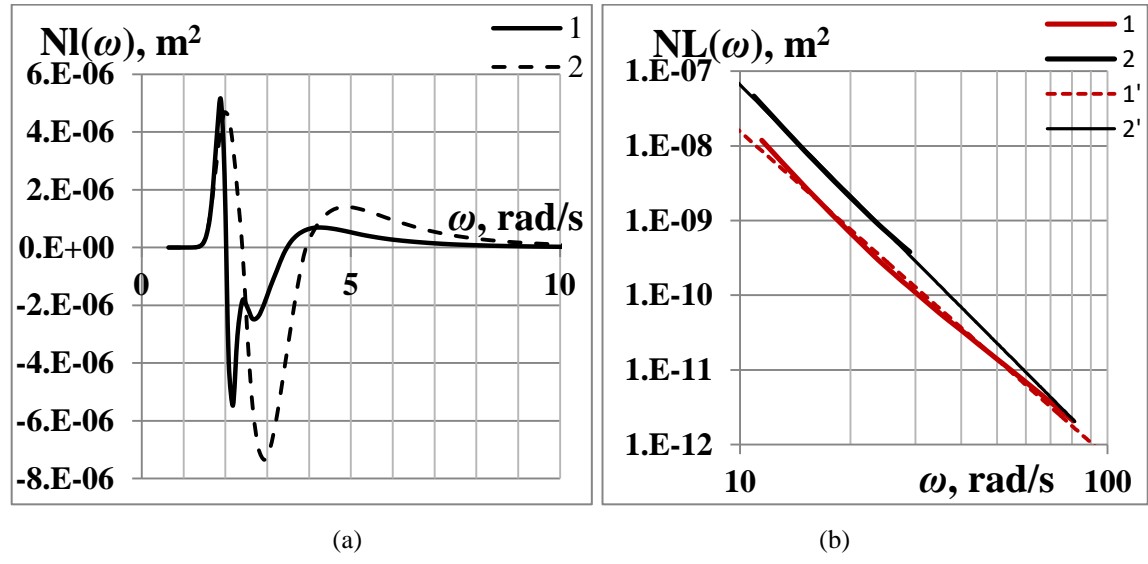

(a)                       (b)

**Figure 1**: (a) Form of one-dimensional NLT, $NL(\omega)$, at low frequencies. Line 1 corresponds to run 1 from table 1, line 2 to run 3 from table 1.

(b) Form of $NL(\omega)$ at high frequencies, run 1 from table 1. Line 1 corresponds to the whole tail of NLT, line 2 to the part of tail ($10 < \omega < 30$) rad/s (with a weight of 3, to separate the lines). Line 1' is the root-mean-square trend of the tail-part 1

15 (equation, $y = 0.0004x^{-4.4}$), and line 2' is the same for the tail-part 2 (equation, $y = 0.002x^{-4.9}$).





The main features of $NL(\omega)$ – an alternating two-lobe shape near the spectral peak and an infinite positive high-frequency tail (Fig. 1a) – are well known (Hasselmann and Hasselmann 1985; Polnikov 1989; van Vledder 2006). Here, it was found that the high-frequency tail of the NLT for all kinds of J-spectrum (14-16) is positive only when n ≥ 4; when n < 4, it is negative and poorly defined due to the weak convergence of the KI for such spectra (Zakharov, 2017). In general, the features of the high-frequency asymptotes for $NL(\omega)$ at the first-step of solving the KE (1) can be formulated as follows:

1) The high-frequency tail of the NLT (for J-spectra) is positive when n ≥ 4 and negative when n < 4;

2) Representation of $NL(\omega)$ in form (18) has a varying power-law of decay;

3) For values $n \geq 5$ , parameter $p$ for $NL(\omega)$ is close to value $n$-1 (i.e., $Nl(\omega) \sim \omega S(\omega)$), in the intermediate frequency region, $5 < \omega/\omega_p < 15$; but in the entire frequency band, $5 < \omega/\omega_p < 25$, the decay of $Nl(\omega)$ is somewhat weaker than $\omega S(\omega)$;

4) For fixed $n > 5$, parameter $p$ increases with a decrease of peakness parameter $\gamma$ and increase of angular anisotropy parameter $a$.

5) When spectrum-decay parameter $n$ approaches to 4, the relative intensity of the NLT-tail decreases radically (Fig. 2a); and when n ≤ 4, the decay-features of NLT begin to depend significantly on the spectrum parameters: $\gamma$, $a$, relative frequency $\omega/\omega_p$ (Fig. 2b), and on the limits of the computing grid in units of $\omega/\omega_p$ . This feature is due to the slow convergence of the KI in such a case (see other numerical details in Polnikov and Uma, 2014).

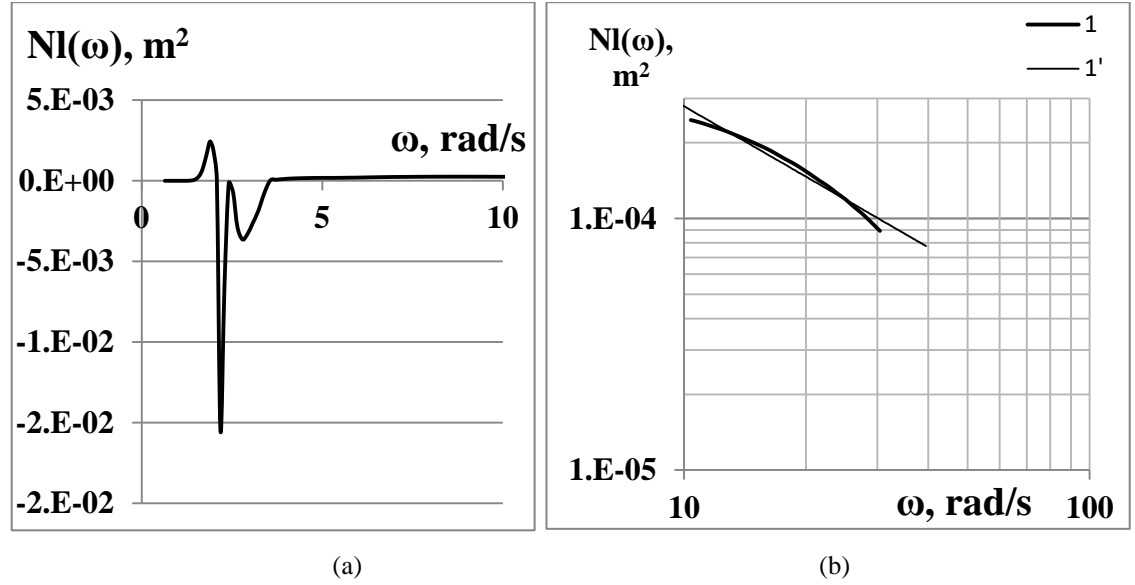

(a)                                                    (b)

**Figure 2**: (a) Form of $NL(\omega)$ at low frequencies for the initial spectrum of run 5 from table 1.
(b) Shape of $NL(\omega)$ at high frequencies for the same spectrum. Line 1 is NLT, line 1' is the root-mean-square trend for the part of tail in the band: (10 < $\omega$ < 30) rad/s (equation, y = 0.0024x$^{-0.9}$).

From the behaviour of the asymptote for NLT, it follows that the nonlinear interactions at high frequencies are sensitive to both the frequency-shape of the spectral peak (i.e., variations of parameter $\gamma$) and the angular shape of the spectrum (i.e.,



variations of parameter *a*) for fixed *n*. Indeed, according to Table 1, despite of the tail of spectrum $S(\omega)$ is the same, the tail of $Nl(\omega)$ is different for different shapes of the spectral peak. It means: *the nonlinear interactions have an explicit non-local feature.* This conclusion has already been noted in literature (Hasselmann and Hasselmann 1985; Polnikov 1989; van Vledder 2006); however, the non-locality of the four-wave nonlinear interactions is expressed in the asymptotes of NLT in the clearest manner. As shown below, the non-locality of NLT is very important in explaining the reasons of the self-similar spectral shape formation.

### 3.2 Kinetic equation solutions on large time scales

To achieve the goals posed, we performed a large series of numerical solutions for KE of type (1) on grid (13) with the algorithm versions both with conservation of total wave action *N* and with retaining total wave energy *E*. The difference between these versions was realised by making the exact balance of positive and negative NLT for either *N* or *E*, respectively, at each time step of the KE numerical solution. For the first time, our calculations have shown that on large scales of evolution time, this difference impacts slightly on the features of the self-similar shapes for $NL(\omega)$ and S$(\omega)$, in the peak-domain ($0.5 \, \omega_p < \omega < 2 \, \omega_p$) , and does not impact on the asymptotes for their high-frequency tails ($\omega > 2 \, \omega_p$). To show these results, we dwell, mainly, on the features for one-dimensional functions.

For each version of the KE-solution, the following results were established:

1) The self-similar one-dimensional spectra, $S_{sf}(\omega)$, in different cases, have slightly different frequency widths *B* and decay with the law

$$S_{sf}(\omega) \sim \omega^{-4 \pm 0.02}; \tag{21}$$

2) The self-similar shapes of $Nl(\omega)$ are slightly different, in different cases, and have the high-frequency asymptotes as

$$Nl(\omega) \sim -\omega^{-4.15 \pm 0.05}; \tag{22}$$

3) The shape parameters for the two-dimensional spectrum $S_{sf}(\omega, \theta)$: $A_p = A(\omega_p)$ and *B*, vary slightly in time within ±5% (as spectra are evaluating), and their average values depend on the degree of anisotropy for the initial spectrum $S(\omega,\theta)$.[1]

The summarised results of calculations for a representative series of initial spectrum shapes $S(\omega,\theta)$ are presented in digital form, for case *E*=const, in Table 2, together with the asymptote parameters for $Nl(\omega)$ and parameters $A_p$ and *B*, for self-similar shape $S_{sf}(\omega, \theta)$. In case *N*=const, values of *B* are on 1-2% smaller, whilst values of $A_p$ are the same (not presented due to similarity). Table 2 shows that the values of $A_p$ and *B* differ significantly in the cases of isotropic and anisotropic initial spectrum $S(\omega,\theta)$, though they have intermediate values (run 7) for weakly anisotropic initial spectra.

---

[1] Hereafter, all estimates of the powers are obtained in the EXCEL-shell with the installed trend-formulas (obtained by the method of least squares). Scattering the exponent powers, in (21), (22) and further, are due to the scattering of the obtained set of estimates, exceeding the statistical errors for the exponents.





Table 2. Asymptote of $Nl(\omega)$ and shape parameters of $S_{sf}(\omega,\theta)$ in the long-term KE solution.

| Run No | Initial spectrum shape | | | Evolution time, s | Asymp. of $Nl(\omega)$ | Parameters of $S_{sf}(\omega,\theta)$ | |
|---|---|---|---|---|---|---|---|
| | $n$ | $\gamma$ | $\Psi(\theta)$ | | $p$ | $B*100$ | $A_p*100$ |
| 1 | 6 | 3.3 | const | $1.3\cdot10^6$ | 4.1 | 22 | 16 |
| 2 | 6 | 1.0 | const | $4.2\cdot10^6$ | 4.2 | 25 | 16 |
| 3 | 5 | 3.3 | const | $1.3\cdot10^5$ | 4.1 | 25 | 16 |
| 4 | 5 | 1.0 | const | $7.9\cdot10^5$ | 4.2 | 25 | 16 |
| 5 | 4 | 3.3 | const | $6.9\cdot10^4$ | 4.1 | 23 | 16 |
| 6 | 4 | 1.0 | const | $5.4\cdot10^4$ | 4.2 | 26 | 16 |
| 7 | 5 | 1.0 | $\cos^2(\theta/2)$ | $4.1\cdot10^6$ | 4.2 | 32 | 46 |
| 8 | 5 | 1.0 | $\cos^8(\theta/2)$ | $3.6\cdot10^6$ | 4.1 | 34 | 63 |
| 9 | 5 | 1.0 | $\cos^2(\theta)$ | $4.1\cdot10^5$ | 4.2 | 33 | 64 |
| 10 | 5 | 1.0 | $\cos^4(\theta)$ | $8.7\cdot10^6$ | 4.2 | 33 | 66 |
| 11 | 5 | 3.3 | $\cos^{12}(\theta)$ | $5.6\cdot10^6$ | 4.2 | 31 | 61 |

Note. Different degrees of shadowing show differences in angular forms for self-similar spectra.
Value $B$=0.33 corresponds to $\gamma$=3.3, $B$=0.63 to $\gamma$=1.0 and $A_p$= 0.16 to $\Psi$= const, $A_p$= 0.64 to $\Psi$= $\cos^2(\theta)$.

5      For anisotropic spectra, the function of angular directivity, $A(\omega)$, includes the sharp peak slightly below peak frequency $\omega_p$ (Fig. 3a). Shape of $A(\omega)$ does not depend of the algorithm version. With the frequency increasing, $A(\omega)$ quickly becomes constant, $A \approx 0.16$ (when $\omega \geq 2\ \omega_p$), corresponding to the isotropic distribution of the spectrum (Fig. 3a). In turn, the frequency shape of one-dimensional spectrum $S_{sf}(\omega)$ has the sharp peak too. For isotropic initial spectra, its shape is very similar to one for the J-spectrum with parameters $\gamma = 3.3$ and $n = 5$ (Fig. 3b). The slight difference of shapes for $S_{sf}(\omega)$, in

10   cases $E$=const and $N$=const, is shown in Fig. 3c, in terms of normalised spectra $S_{nor}(\omega/\omega_p)$. For the anisotropic initial spectra, the shape of peak for $S_{sf}(\omega)$ is similar to that above, although the peak is 1.5 times wider (in the value of $B$). Such a detailed description of the integral parameters for two-dimensional self-similar spectrum $S_{sf}(\omega, \theta)$ essentially supplements the results of earlier works (Polnikov 1990; Pushkarev et al. 2003; Badulin et al. 2005). It has a reasonable academic interest.

     The shapes of self-similar one-dimensional functions $Nl(\omega)$. are shown in Figs 4a,b in the normalized form, $Nl_{nor}(\omega/\omega_p)$

15   for several time moments, It is seen that $Nl(\omega)$ has only two lobes, which differ slightly for cases of $E$=const. and $N$=const, providing the proper balances (11). In terms of Q (Eq. 17), they are about 0.05-0.1%. More details about features of $Nl_{nor}(\omega/\omega_p)$ are out of this paper aims. The most important fact about shapes of $Nl(\omega)$ for the below analysis is that, in any case, their tail asymptotes have form (22).



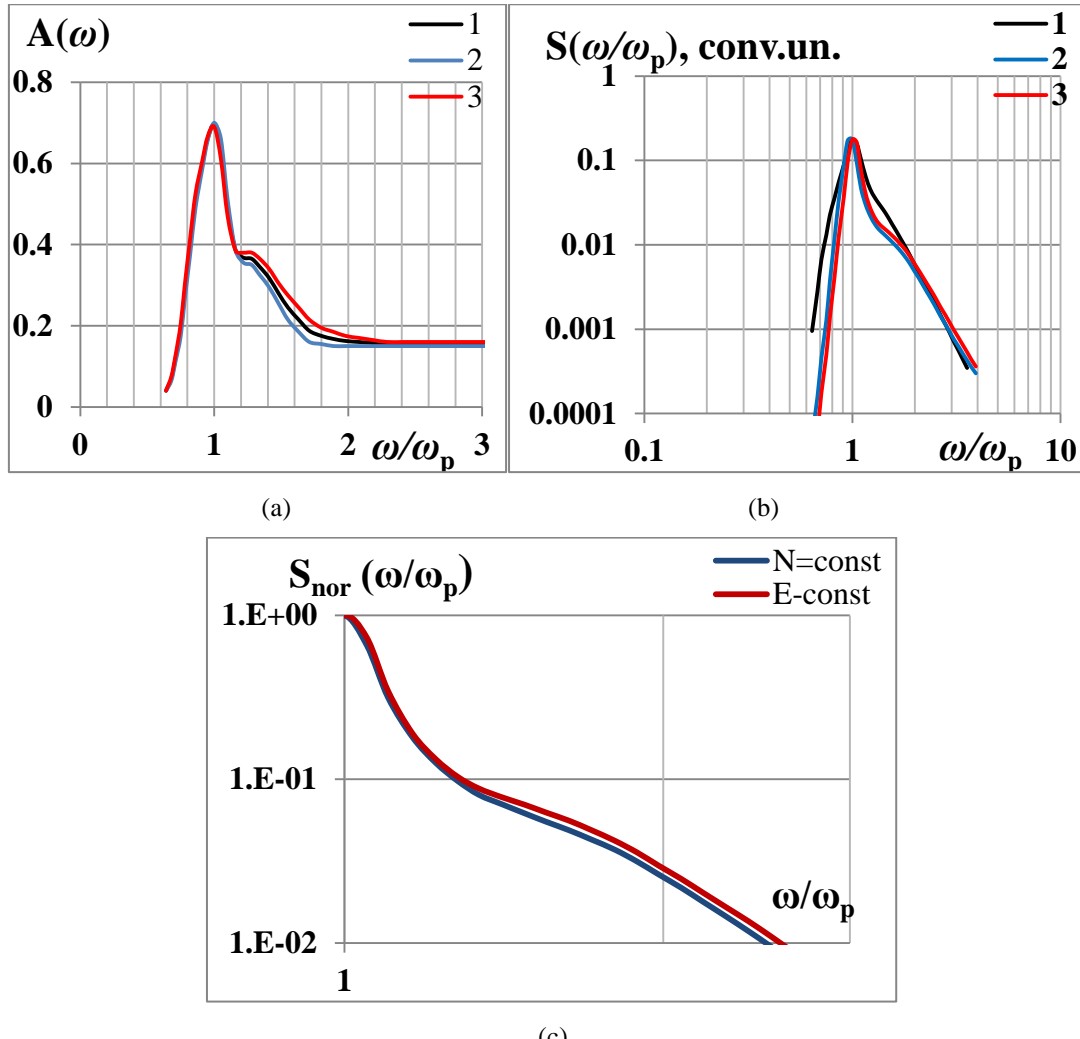

(a)                                            (b)

(c)

**Figure 3**: (a) Typical shape of the self-similar function for the angular directivity, $A(\omega)$. Line 1 is the mean for all anisotropic runs (7 through 11) from Table 2; line 2 is run 10; line 3 is run 11.
(b) Self-similar shapes $S_{sf}(\omega/\omega_p)$ for run 3 from Table 2 (isotropic spectrum). Line 1 is the shape of J-spectrum according to (14-16) (the same run); line 2 is $S_{sf}(\omega/\omega_p)$ at time t ≈ $1 \cdot 10^4$ s; line 3 is $S_{sf}(\omega/\omega_p)$ at the time t = $1 \cdot 10^5$ s.
(c) The difference of shapes for normalised self-similar spectra $S_{nor}(\omega/\omega_p)$ in two cases. Run 4 from Table 2.
                                            .

Detailed information regarding the features for spectrum evolution is shown in Figs. 5-8 (using the versions of KE-solution with constant wave action $N$ are marked in the proper captions). The time-history of a one-dimensional spectrum $S(\omega)$ for run 1 from Table 2 is presented in Fig. 5a. The corresponding time-history of one-dimensional NLP $Nl(\omega)$ is shown in Fig. 5b. It can be seen that when the self-similar shape of the spectrum is practically formed (at a time of the order of $4 \cdot 10^4$ s, or more), the NLT changes principally in such a way that the intensity of the tail for $Nl(\omega)$ becomes fast going down. Pay attention that for $\omega > 2\omega_p$, $Nl(\omega)$ is never equal to 0, and its asymptotic behaviour is defined by formula (22). The same situation is realised for all the cases considered (with small variations for parameter $p$, as shown in Table 2).




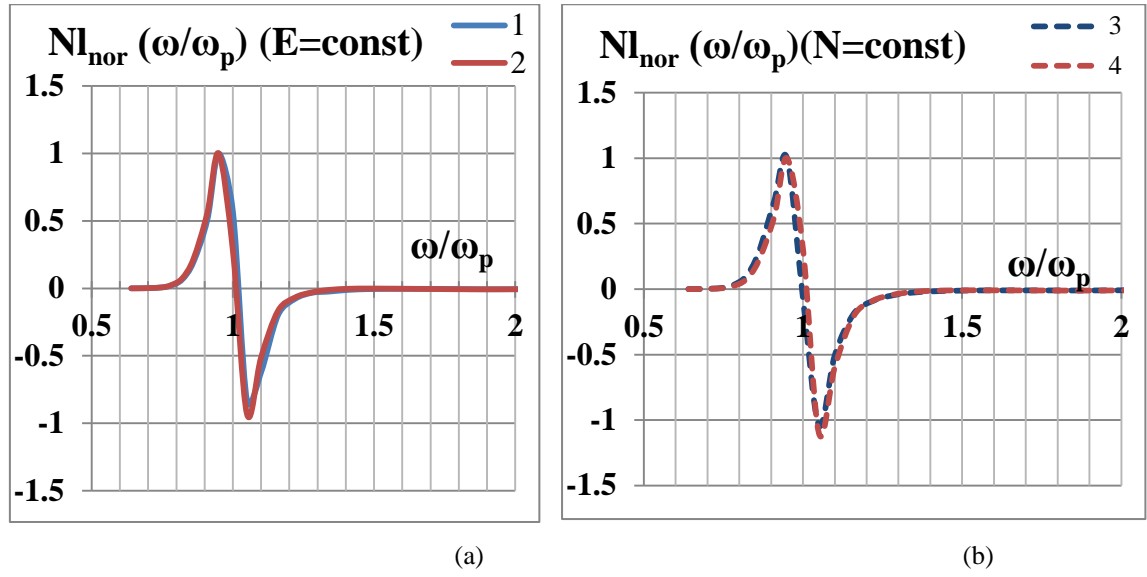

(a)                                                   (b)

**Figure 4**: The self-consistent one-dimensional normalised functions, $Nl_{nor}(\omega/\omega_p)$. Run 4 from Table 2.
(a) The case of preservation for $E$; (b) The case of preservation for $N$;
Line 1 corresponds to t = $1.0 \cdot 10^5$ s; line 2 to $3.5 \cdot 10^5$ s; line 3 to $1.8 \cdot 10^5$ s; line 4 to $6.1 \cdot 10^5$ s.

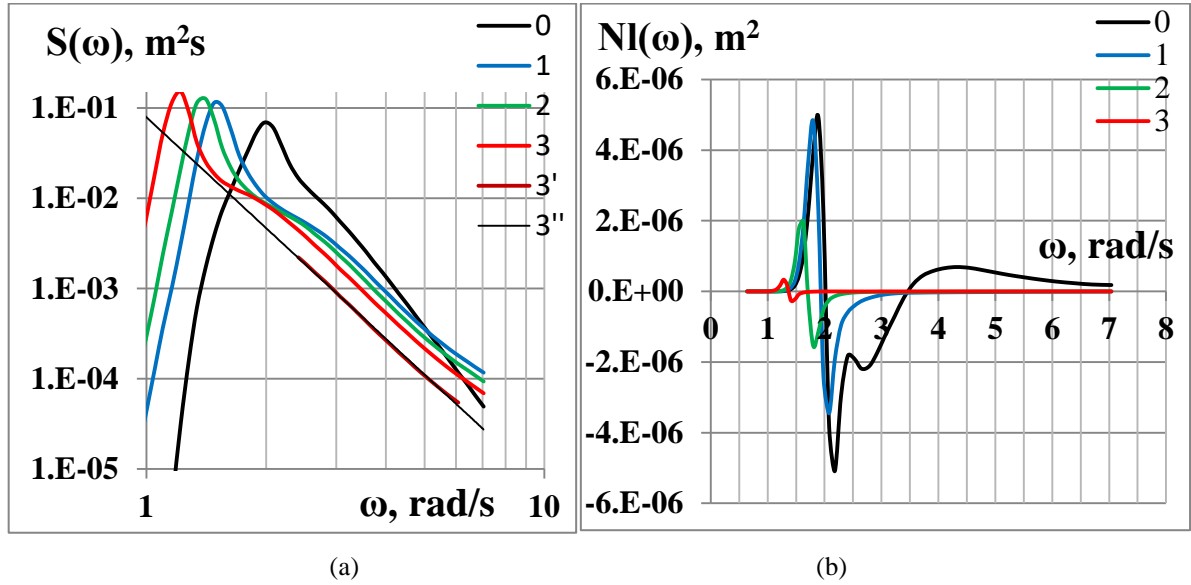

(a)                                                   (b)

**Figure 5**: (a) Time-history of one-dimensional spectrum $S(\omega, t)$ for run 1 from Table 2. Line 0 corresponds to $t = 0$ s; line 1
to $t = 1.4 \cdot 10^5$ s; line 2 to $t = 3.0 \cdot 10^5$ s; line 3 to t = $1.2 \cdot 10^6$ s; line 3' is the tail part of line 3 (with a weight of 0.3); 3" is the
trend of tail 3' (equation y = $0.0792x^{-4.1}$).
(b) Time-history of one-dimensional NLT $Nl(\omega, t)$ (the same run). Line 0 corresponds to $t = 0$ s; line 1 to $t = 470$ s; line 2 to
$t = 4.5 \cdot 10^4$ s; line 3 to $t = 3.7 \cdot 10^4$ s.





In this respect, run 6 from Table 2 is very instructive (Figs. 6a, b). Figure 6a demonstrates the evolution of the initially isotropic spectrum ($a = 0$) with parameters $\gamma = 1$ and $n = 4$. In this case, for the initial J-spectrum decaying with the law: $S(\omega) \sim \omega^{-4}$, Figure 6b shows that, at the first time-step, the intensity of the tail for nonlinear transfer $Nl(\omega)$ is not small, as it might

be expected in accordance with the conclusions by Zakharov and Filonenko (1966). Later, on the relatively great evolution scales, $t > 10^4 \cdot 1/\omega_p(0)$, the self-similar spectrum has already formed, having the same decay-law as the initial spectrum: i.e., $S(\omega) \sim \omega^{-4}$. Though, on these scales, the nonlinear transfer gets the rapidly decaying tail of form (22), and the intensity of this NLT-tail is already very small, corresponding to the theory of Zakharov and Filonenko (1966). The same results we have got using initial spectra with n < 4 (e.g., the case with n = 3.5, which is not shown, as it is very similar to run 6, and

Figs. 6a,b). The said is very interesting and important result shown here for the first time.

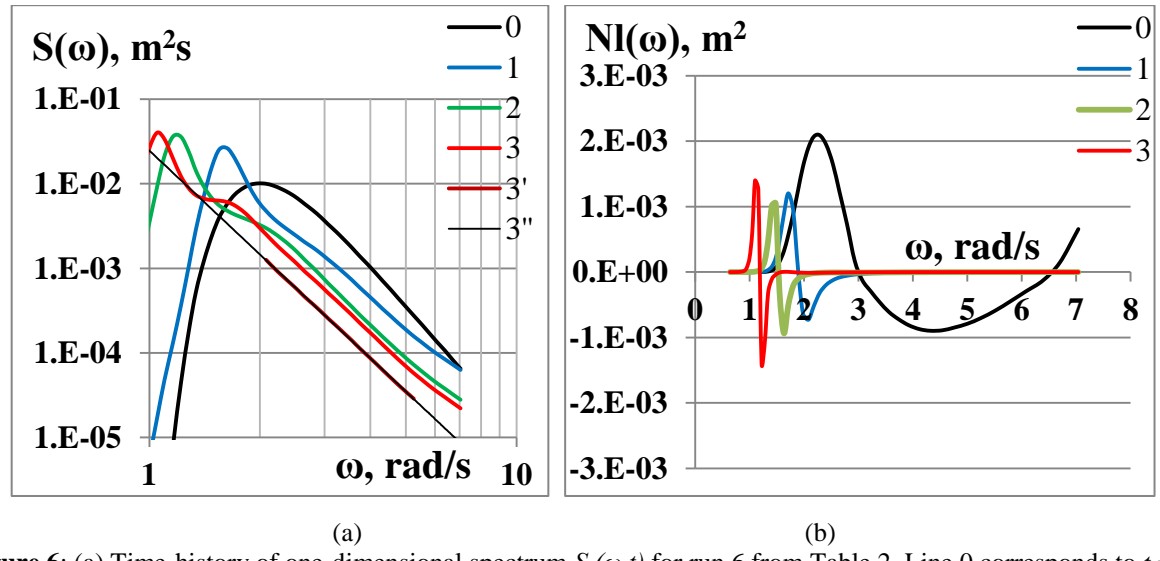

(a)                                                                    (b)

**Figure 6**: (a) Time-history of one-dimensional spectrum $S(\omega,t)$ for run 6 from Table 2. Line 0 corresponds to $t = 0$ s; line 1

to $t = 5.7 \cdot 10^2$ s; line 2 to t $= 3.9 \cdot 10^3$ s; line 3 to t $= 5.4 \cdot 10^4$ s; line 3' is the tail part of line 3 (with a weight of 0.3); 3" is the trend of tail 3' (equation y = 0.0248x$^{-4.1}$).

(b) Time-history of one-dimensional NLT $Nl(\omega, t)$ (the same run). Line 0 corresponds to $t = 0$ s with a weight of 1); line 1 to $t = 570$ s (with a weight of 3); line 2 to $t = 3.9 \cdot 10^3$ s (with a weight of 10); line 3 to $t = 5.4 \cdot 10^4$ s (with a weight of 100).

Undoubtedly, the main reason for the small values of the tail-intensity for function $Nl(\omega)$ at a large $t$ is the formation of the specific, self-similar shape $S_{sf}(\omega,\theta)$ for the entire spectrum. In this case (run 6), the main difference between initial form $S(\omega,\theta)$ and the final one, $S_{sf}(\omega,\theta)$, is determined only by the self-similar frequency-shape of the peak-domain for $S_{sf}(\omega,\theta)$, because the angular distribution of the spectrum is always isotropic. This fact testifies, first, the nonlocality of the four-wave nonlinear interactions (already mentioned above) and, second, the fundamental role of the spectrum-peak shape in forming

the tail of NLT-function $Nl(\omega)$ with the frequency-asymptote of form (22).



Here, it is important to emphasise that the results presented above were repeated for versions of solutions for KE (1) both with preservation of total wave energy $E$ and total wave action $N$ too (e.g., Figs. 7a,b below). Moreover, the same results were found in the algorithms without the exact preservation of $N$ or $E$, when the proper control was not carried out.

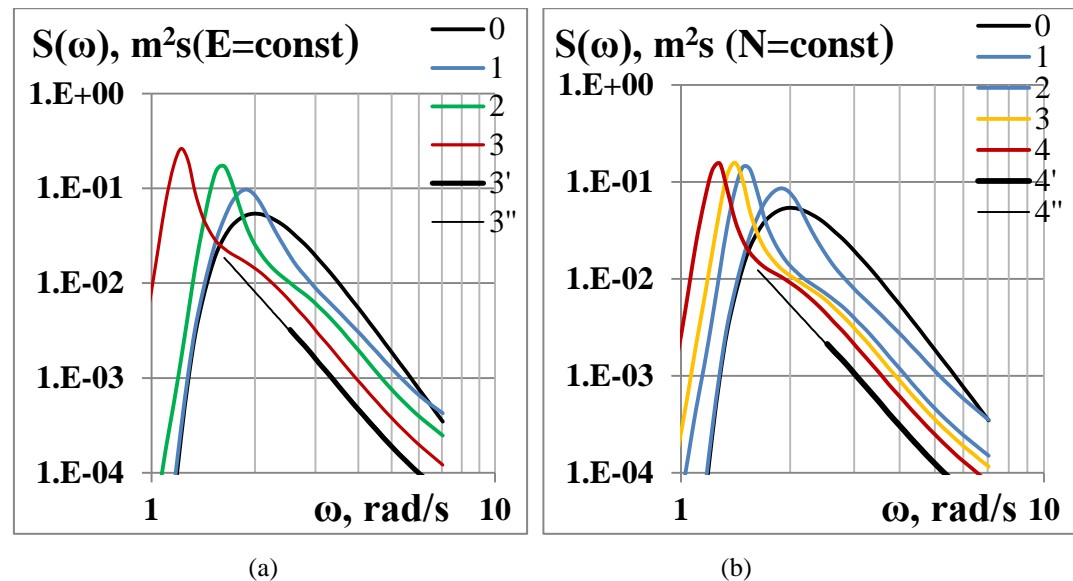

(a)                 (b)

**Figure** 7: Time-history $S(\omega,t)$ for two versions of the KE-solution (run 4):
(a) The version with condition $E$ = const. Line 0 corresponds to $t = 0$ s; line 1 to $t = 1830$ s; line 2 to $t = 0.5 \cdot 10^5$ s, line 3 to $t = 3.5 \cdot 10^5$ s, line 3' to the tail part of line 3 (with a weight of 0.5), line 3'' to the trend for line 3' (equation y= 0,135·x$^{-4.06}$).
(b) The version with condition $N$ = const. Line 0 corresponds to $t = 0$ s; line 1 to $t = 2500$ s; line 2 to $t = 6.7 \cdot 10^4$ s, line 3 to $t = 1.7 \cdot 10^5$ s, line 4 to $t = 6.1 \cdot 10^5$ s, line 4' to the tail part of line 4 (with a weight of 0.5), line 4'' to the trend for line 4' ( equation     y= 0,09·x$^{-4.06}$)

At the same time, some numerical values are different for the various versions of the algorithms, as seen from the temporal asymptote for certain spectral parameters. It is evidently true for total wave action $N(t)$ and total wave energy $E(t)$. The same can be said about downshifting the peak frequency, $\omega_p(t)$ (see below). But more interesting is the behaviour of the fluxes for wave energy $P_E$ and wave action $P_N$, taking place in two different versions for the KE-solution algorithms.

To see this, the frequency functions of fluxes for wave energy $P_E$ and wave action $P_N$ were estimated with the formulas similar to ones from (Pushkarev et al. 2003; Badulin et al. 2005):

$$P_E(\omega) = -\int_{\omega_1}^{\omega} \int Nl(\omega,\theta)\,d\theta\,d\omega \quad \text{and} \quad P_N(\omega) = -\int_{\omega_1}^{\omega} \int (Nl(\omega,\theta)/\omega)\,d\theta\,d\omega, \tag{23}$$

where $\omega_1$ is the lower limit of the numerical frequency band. According to (Pushkarev et al. 2003; Badulin et al. 2005), the positive value of $P_E$ (or $P_N$) indicates the flux upward in frequency and the negative value does downward.

For run 2 from Table 2, the time-history of fluxes $P_E(\omega,t)$ and $P_N(\omega,t)$, under the condition of constant wave action $N$, is presented in Figs. 8a,b, and in Figs. 9a,b, under the condition of constant wave energy $E$. Figures 8a,b show that if the wave

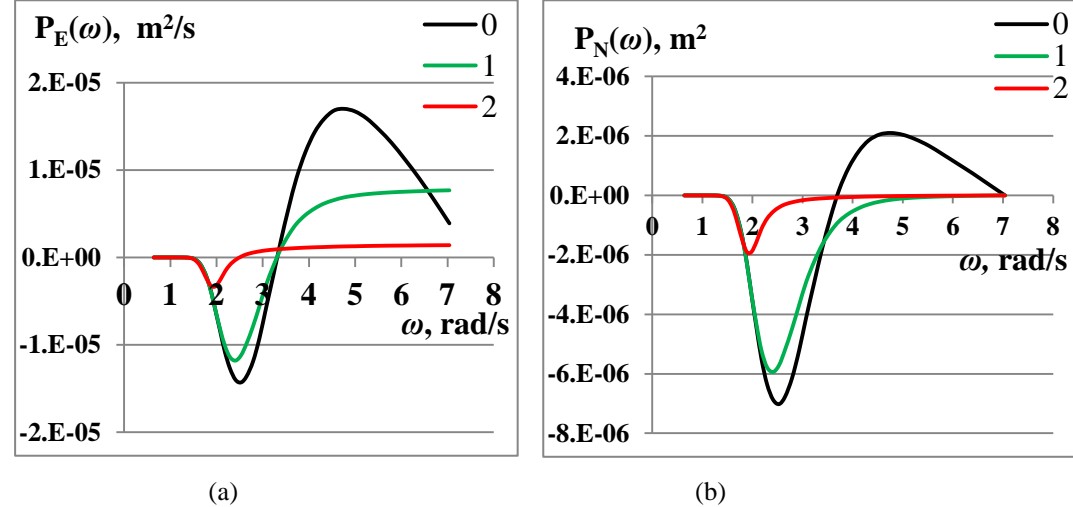

**Figure 8**: (a) Time-history of flux $P_E(\omega,t)$ for run 2 from table 2 (when $N$ is constant). Line 0 corresponds to $t = 0$ s; line 1 to $t = 700$ s; line 2 to $t = 2500$ s.

(b) Time-history of flux $P_N(\omega,t)$ for the same run 2. Lines correspond to the same time moments.

action $N$ is constant, a quasi-constant tail for upward energy flux $P_E(\omega)$ is established in frequencies range $\omega > 2\omega_p(t)$, on long evolution scales: $t > 10^4/\omega_p(0)$. The latter means the loss of wave energy in the system. The flux at the upper edge of frequency band, $P_N(\omega_{max},t)$, is always zero, what means preservation of $N$. Under the condition of constant wave energy $E$, the same is true for downward flux $P_N(\omega)$ (Figs. 9a, b). Pay attention that, in proper cases, the tails for both fluxes are not constant in $\omega$ but to quasi-constant, because of $Nl(\omega)$ is not equal to 0 at higher frequencies. The fluxes asymptotes are defined by formulas (22, 23). Due to very week dependences of these tails for $P_E(\omega)$ and $P_N(\omega)$ on $\omega$, we could call them as the constant ones, following to (Pushkarev et al. 2003; Badulin et al. 2005) for simplicity of the further analysis.

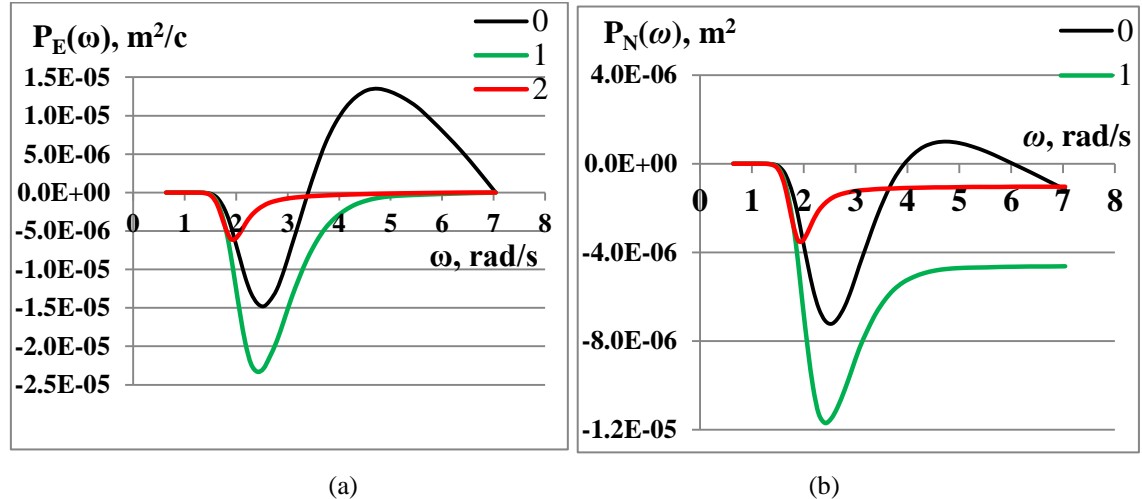

**Figure 9**: (a) Time-history of flux $P_E(\omega,t)$ for run 2 from table 2 (when $E$ is constant). Line 0 corresponds to $t = 0$ s; line 1 to $t = 250$ s; line 2 to $t = 1830$ s.

(b) Time-history of flux $P_N(\omega,t)$ for the same run 2. Lines correspond to the same time moments.



Herewith, the shape of the self-similar spectra obtained in both versions of the KE-solution is nearly the same, though, the spectra differ in their intensities (Figs. 7a,b) and in the rate of formation, defined by different versions of the KE-solution algorithms (see below). This difference is manifested via the different temporal asymptotes for peak frequency $\omega_p(t)$, values

of spectral peak, $S_p(t) = S_{sf}(\omega_p)$, and, evidently, temporal asymptotes for integral characteristics: $E(t)$ or $N(t)$ (in proper case). All of these asymptotes could be derived analytically, among them the basic one is the temporal downshifting of peak frequency, $\omega_p(t)$.

### 3.3 Correspondence between numerical and analytical results

Basing on the self-similarity for $S_{sf}(\omega,\theta)$, one can analytically show that for constant $N$, it should takes place the asymptote (Pushkarev et al. 2003; Badulin et al. 2005)

$$\omega_p(t) \propto t^{-1/11} \qquad , \tag{24}$$

whilst, for constant $E$, it is true (not presented in literature)

$$\omega_p(t) \propto t^{-1/9} \qquad . \tag{25}$$

Details of derivations for ratios (24, 25) can be found in the Appendix (A) of this paper. But here we dwell on manifestations of differences in asymptotes for $E(t)$, $N(t)$, and $S_p(t)$ in different version of the KE-solution algorithms, attracting proper our results for isotropic initial spectra.

First, let us consider the case of total wave energy $E$ preservation (as the less studied). If the self-similar one-dimensional spectrum, $S_{sf}(\omega)$, is represented by ratio (9), then we have

$$E(t) = \int S_{sf}(\omega,\omega_p(t))d\omega = \int S_{sf}(\omega_p)F_{sf}(\omega/\omega_p)d\omega \propto S_{sf}(\omega_p)\cdot\omega_p(t) = \text{const.} \tag{26}$$

Consequently, the peak of spectrum, $S_p(t)$, should grow as

$$S_p(t) \sim 1/\omega_p(t), \tag{27}$$

and the total wave action should grow at the same rate:

$$N(t) = E(t)/\omega_p(t) \propto 1/\omega_p(t) \qquad . \tag{28}$$

(This growth of $N$ will be discussed below). Our calculations show that under the condition of constant wave energy, the peak frequency of the self-similar spectrum decreases according to the law

$$\omega_p(t) \sim t^{-0.09 \pm 0.02}. \tag{29}$$

Within the limits of the numerical estimates scattering, ratio (29) corresponds to (25) rather well. Herewith, the numerical asymptotes for growing values $S_p(t)$ and wave action $N(t)$ are as follows

$$S_p(t) \sim t^{0.08 \pm 0.02} \quad \text{and} \qquad N(t) \sim t^{0.08 \pm 0.02}, \tag{30}$$

what corresponds to the basic dependence (29) and ratios (27, 28) very well.



Under the condition of constant wave action $N$, ratio (26) should be replaced by the following one

$$N(t) = \int \left[ S_{sf}(\omega,t) / \omega \right] d\omega \sim E(t) / \omega_p(t) = \text{const.} \tag{31}$$

Hence, total energy $E(t)$ should decrease as

$$E(t) \sim \omega_p(t) \tag{32}$$

(this leakage of energy will be discussed below), and the ratio

$$S_p(t) = \text{const} \tag{33}$$

must take place.. Under the condition of constant wave action $N$, our calculations give the dependence

$$\omega_p(t) \sim t^{-0.075 \pm 0.02}. \tag{34}$$

As seen, dependence (34) is remarkably weaker then (29), what is in full accordance with the analytics. Despite of ratio (34)

is slightly weaker than the theoretical dependence, within the limits of estimates scattering, it is rather close to (24). Herewith, in this case, our calculations show that

$$E(t) \sim t^{-0.075 \pm 0.02}, \tag{35}$$

and

$$S_p(t) = \text{const}, \tag{36}$$

what means that the expected theoretical dependences (32, 33) are very well fulfilled.

Thus, in general, the correspondence between numerical and analytical results is rather well. The mentioned deviations of numerical ratios (34) and (29) for $\omega_p(t)$ from the analytical ones,(24) and (25), are apparently due to the limited range for variability of values $\omega_p$ in our calculations, determined by grid (13), and to relatively small scales of the considered evolution time. The sophisticated calculation could be executed in a separate study with using the properly chosen grid.

According to theoretical estimates (27) and (33), the only visually observable effect of the differences in these versions of the KE-calculation algorithms is the different asymptotes of the spectrum peak $S_p(t)$. This effect is really observed in our calculations (Figs. 7a,b).

In addition to the said, it is interesting to give estimates of the self-similar spectrum values at any fixed frequency $\omega_{fix}$ in the tail region. They are determined by the ratios (for any version of the algorithm for the KE-solution)

$$S_{sf}(\omega_{fix}) = S_{sf}(\omega_p) \cdot F_{sf}(\omega_{fix} / \omega_p) \propto S_{sf}(\omega_p) \cdot (\omega_{fix} / \omega_p)^{-4}, \tag{37}$$

where the tail for self-similar shape-function $F_{sf}(\omega / \omega_p)$ is explicitly written. In the case of constant $E$, from (37) and (27), the following dependence should take place

$$S_{sfE}(\omega_{fix}) \propto (\omega_p)^{-1} \omega_{fix}^{-4} \omega_p^4 \propto \omega_p^3. \tag{38}$$

For constant $N$, according to (36, 37), it must be valid



$$S_{sfN}(\omega_{fix}) \propto \omega_p^{4}. \tag{39}$$

Our calculations give the following estimates. For constant $E$, we have

$$S_{sfE}(\omega_{fix}) \sim t^{-0.24 \pm 0.01}, \tag{40}$$

and for constant $N$, it reads

$$S_{sfN}(\omega_{fix}) \sim t^{-0.29 \pm 0.01}. \tag{41}$$

Numerical dependences (40, 41) correspond very well to the basic numerical dependences for $\omega_p(t)$: (29) and (34), if one accounts for the above scattering in the estimates of powers. Pay attention that in the case of constant $N$, the intensity of spectral tail decreases faster, to support the proper self-similar spectral shape.

These estimates complete our description of new peculiarities for the numerical solution of KE, established in this work.

## 4 Interpretation of results

The peculiarities of the frequency asymptote of NLT, $Nl(\omega)$, and the features of the self-similar spectrum formation (Sects. 3.1, 3.2), which are accompanied by the peak-frequency asymptotes (24) and (25) and related asymptotes (32,33) and (27,28) (Sect. 3.3), do not require any special treatment. The only principal point for treatment is the search for reasons of establishing self-similar spectrum $S_{sf}(\omega)$ with the tail of form (21), because it is realised for any version of the algorithm for solution of KE (1). Although this numerical result, $S_{sf}(\omega) \sim \omega^{-4}$, was obtained earlier under the condition of wave-action constancy (Pushkarev et al. 2003; Badulin et al. 2005), here the question arises: why the one-dimensional self-similar spectrum, $S_{sf}(\omega)$, always has the tail of form (21), both in cases of preservation of the total wave action and in cases of conservation of the total wave energy. An interpretation of this result is given below.

First, we note that the possibility of separating these algorithms is stipulated by the fact that two integral parameters of the wave-system, $N$ and $E$, are not simultaneously conserved in the numerical solution of KE (1). This statement was repeatedly noted by Zakharov and co-authors (e.g., Pushkarev and Zakharov, 2000; Pushkarev et al., 2003; Badulin et al., 2005; Zakharov 2017, and references therein). The absence of the simultaneous preservation of $N$ and $E$ for a self-similar shape solution of KE (1) was discussed in Sect. 2. Here we may note some technical and mathematical reasons for non-presentation of values for $N$ or $E$ in the course of the KE numerical solution. The former include: using restricted numerical grid, systematic errors in the calculation of KI, provided by approximate estimations of contributions from the singular points in the integrand (Polnikov 1989, van Vledder 2006), and insufficient resolution. The mathematical reasons are: the weak convergence of the KI for slow-falling spectra (Zakharov 2017) and typical shortages of numerical solution for the KE. These reasons are not due to the physics of the phenomenon, because KE of form (1) has no external sources or sinks, which could violate the constancy of $N$ or $E$. Therefore, from the viewpoint of mathematics and physics for the system under study, both versions of the computing algorithm for KE (1) have a reason to be used, at least as a compensating the mentioned errors.





As shown above, the differences among these versions of the KE solutions lead to fundamental differences in the formation of fluxes for wave action $N$ and wave energy $E$. When $N$ is constant, on the scales of long-term evolution, the (quasi-)constant upward energy flux $P_E(\omega)$ is realised in the tail frequency-range. In turn, if $E$ is constant, the similar downward action flux $P_N(\omega)$ occurs (Figs. 8a and 9b). Thus, two fundamentally different physical situations arise, provided

by the choosing the algorithm version. Nevertheless, the tail of one-dimensional self-similar spectrum, $S_{sf}(\omega,)$, always takes form (21) (Figs. 7a,b). The interpretation of this result requires a separate discussion, and, may be, a separate research.

Indeed, according to Zakharov and Zaslavskii (1982), the spectra of forms (6) and (21) are treated as the Kolmogorov-type spectra of constant energy flux $P_E(\omega)$ directed upward in frequencies. Therefore, they must be formed only in the version of the KE solution with the condition of total wave action preservation, when proper flux $P_E(\omega)$ occurs. According to

this logic, in the case of the KE-solution with the condition of constant wave energy, the spectra with the tail of form (7) should be formed, that is, the tail should take the form $S_{sf}(\omega) \sim \omega^{-11/3}$. If it is so, there are not any problems, and the treatment by Zakharov and Zaslavskii (1982) could be accepted totally.

However, the spectrum tail of form $S_{sf}(\omega) \sim \omega^{-11/3}$ is not observed in our KE-solutions with the version of constant $E$. This result is not related to the accuracy of the calculations, because the estimated numerical errors of our calculations are

only 3% to 5% (Sect. 2.1), and the errors of estimating the exponents in the asymptotes are not more than 1% to 2%. That gives grounds for distinguishing the '-4' and '-11/3' decay laws. Therefore, the numerical result of invariability of the solution of form (21) should be considered as the quite reliable one.

Our treatment of the matter under discussion is based on the results of estimating the high-frequency asymptote for nonlinear energy transfer $Nl(\omega)$, obtained above (Sect. 3.1). Indeed, by setting up a standard initial spectrum of waves in

form (14-16) with the decay index n > 4, we found that, for such $n$, the tail of $Nl(\omega)$ is always positive and decays rapidly (see Figs. 1a, 2a and the asymptotes in Table 1). Consequently, in the course of spectrum evolution in terms of KE (1), at the beginning stage, the intensity of the spectrum tail increases, resulting in decreasing the decay-power of current spectrum. On the time scales when the tail of the spectrum approaches to form $S(\omega) \sim \omega^{-4}$, the asymptote of $Nl(\omega)$ has already changed its sign and took form (23), i.e., close to $Nl(\omega) \sim -\omega^{-4}$. Mathematically, it is evident that this change of $Nl(\omega)$ stabilises the

spectral shape, providing the main reason of establishing self-similarity of the spectral shape with the tail of form $S_{sf}(\omega) \sim \omega^{-4}$. Pay attention that a choosing the algorithm version does not impact to this fact, as is dose not impact of the tail of $Nl(\omega)$.

It should be noted here that, due to the nonlocality of four-wave nonlinear interactions, the asymptote of $Nl(\omega)$ is determined not only by the asymptotic behaviour of the spectrum tail but does also by the shape of its peak. The latter is clearly seen from the results of the calculations for run 6 from Table 2, when the initial spectrum is falling with the '-4' law

(Figs. 6a,b). This manifests the role of the spectrum-peak domain in formation of the entire shape of self-similar spectrum $S_{sf}(\omega,\theta)$. Thus, due to reforming both the shape of spectral peak domain and the spectrum tail, in the course of the KE-solution, the coincidence of power-forms for asymptotes for $S(\omega)$ and $Nl(\omega)$ is realised, which stabilises the shape of the entire spectrum, providing the tail of form (21). Moreover, our numerical experiments for initial spectra with n < 4 showed that, although the KI is not well definite for such spectra, the stabilising process is even faster, because of the tail of $Nl(\omega)$, in





such cases, has a negative sign from the very beginning of spectrum evolution. Due to the non-locality of nonlinear interactions, the change in the spectral peak results in the radical change in the convergence of the KI, improving it. Eventually, on the scales of long-term evolution, there is no problem with the convergence of the KI (Figs. 4a,b, 5b, 6b).

Basing on the said, we assume that the establishing the universal form of the self-similar spectrum is due only to the
mathematical feature of the KI itself. This feature has such a nature that, despite of version for solution algorithm, the self-similar spectral shape is restored very quickly for initial spectra with any decay law, resulting in the described form for $S_{sf}(\omega,\theta)$ with tail (21). This new mathematical fact indicates a high degree of self-organisation of the nonlinear processes under consideration. The said allows us to conclude that the formation of a self-similar spectrum with tail $S(\omega)\sim\omega^{-4}$ is not connected with the constant energy or action fluxes, which are formally realised in the numerical solution for KE (1).

Nevertheless, the (quasi-) constant fluxes are present at the tail frequency range. Thus, it is important to clarify the point: what is the reason for their existence? In other words: what is the reason of the energy leakage (for constant $N$) and wave action growing (for constant $E$)?

In view of the absence of any external sources and sinks, we can postulate that the self-similar frequency-shape of the spectral peak, located in domain $0.8\omega_p \leq \omega \leq 2\omega_p$, plays the proper role of the internal sink or source in this system, whilst
the spectral tail range, $\omega \geq 2\omega_p$, plays the counterpartying role. Herewith, in the upper frequency range, $\omega \geq 2\omega_p$, the peak-domain is followed by a sharp transition in angular function $\Psi(\omega,\theta)$, resulting in an isotropically distributed power-law tail of frequency-form (21) (Fig. 3a). Eventually, the self-similar spectrum with the tail of form $S(\omega)\sim\omega^{-4}$ is stipulated only by the mathematical properties of the KI, which were established and proved analytically by Zakharov and Filonenko (1966) (in the part of principal possibility of existing the spectral solutions for KE (1) of form $S(\omega)\sim\omega^{-4}$).

Summarising this interpretation of the obtained results, one can state that we do not deal with the Kolmogorov-like turbulence in the frame of KE (1). Such a conclusion could be justified by three reasons: a) the presence of a distinguished frequency scale in the system (provided by the peak of the spectrum); b) the nonlocality of four-wave nonlinear interactions formatting the entire spectrum; and c) the absence of sources and sinks external to the waves. In this case, the observed ranges of the (quasi-) constant fluxes $P_E(\omega)$ and $P_N(\omega)$ and the leakage of total energy $E$ with asymptote (32) or the growth
of action $N$ in form (28) are merely the formal mathematical consequences of the accepted versions of the computational algorithm for solving KE (1). As far as the final spectrum-tail shape is independent of fluxes $P_E$ and $P_N$, they cannot be recognized as the reason of establishing the self-similar spectrum with the tail of form (21).

In addition, it is pertinent to note here that, in this problem, using the condition for wave energy constancy is preferable, from the physical point of view. It is because of the constancy of total energy $E$ is proved (Zakharov, 1968), whilst total
action $N$ is only an auxiliary analytical variable, the invariance of which does not follow directly from the Euler equations. There are no physical reasons for the leakage of total energy in the conservative system considered. Note that the fact of realization one of these cases can be validated via empirical checking the presence of asymptotes (27) or (3a) in proper wave-tank experiments. By the way, some evidence of realization for asymptote (27) can be found in (Shugan et al., 2014).





In turn, according to the theoretical point of view (Zakharov at al., 1992), the wave action has a formal meaning of virtual "wave particles". Thus, the fact of increasing $N$, in the case of constant energy of the system, could be interpreted as the formal effect of 'condensation' of wave-particles from the tail of the spectrum during the nonlinear wave evolution in the frame of KE (1).

Another situation, relevant to the Kolmogorov-type turbulence, is realised in the case of presence of source and sink external to the wave system, separated in the frequency band. It was modelled in numerous works devoted to numerical solution of the KE of form (8) (e.g., Polnikov 1990, 1994; Pushkarev et al. 2003; Badulin et al. 2005; among others). However, the detail discussion of the impact of the mentioned algorithms on process for Kolmogorov-like spectra formation in forms (6) and (7), detailed analysis of these processes, and their reconciliation with the conclusions of this paper requires a

further study and separate presentation in future.

## 5 Conclusions

Summarising the results above, we can draw the following conclusions.

First, the features of the high-frequency asymptotes for nonlinear energy transfer $Nl(\omega)$, which are close to a power-law

form, play a fundamentally important role in understanding the evolution features of nonlinear gravity waves in water, governed by the four-wave KE of form (1). In particular, the behaviour of these asymptotes indicates the fact of nonlocality of nonlinear interactions under consideration. The change of the sign for $Nl(\omega)$ and its asymptote to form $Nl(\omega) \sim -\omega^{-4}$, associated with a change in the decay-law of spectrum tail to form $S(\omega) \sim \omega^{-4}$, is responsible for maintaining the tail of spectral shape in form $S_{sf}(\omega) \sim \omega^{-4}$.

Second, the shape of the peak-domain for two-dimensional spectrum $S(\omega,\theta)$, the characteristics of which are presented in Table 2 and in Figs. 3a, b, c, is responsible for the formation of self-similar two-dimensional spectrum $S_{sf}(\omega,\theta)$ as a whole, what is clearly visible on the example of calculations for run 6 from Table 2, for the initial spectrum with $n = 4$ (Figs. 6a, b).

Third, self-similar two-dimensional spectrum $S_{sf}(\omega,\theta)$ with the tail of form $S_{sf}(\omega) \sim \omega^{-4}$, supported by a nonlinear energy transfer with the tail of form $Nl(\omega) \sim \omega^{-4.15 \pm 0.05}$, is established in solutions of KE (1) for any initial spectral forms (14-16), in

the versions of both preservation of total wave energy $E$ and constancy for total wave action $N$ (Figs. 7a,b). As a result, the establishing self-similar spectra with the tail $S_{sf}(\omega) \sim \omega^{-4}$ is the purely mathematical property of the four-wave KI, that was analytically found and proved by Zakharov and Filonenko (1996).

The (quasi-) constant downward fluxes of wave action $P_N(\omega)$ and the upward energy fluxes $P_E(\omega)$, taking place in the spectrum-tail range, are not the reasons for self-similar spectrum formation in the course of solving KE (1), because of the

deviation of this situation from the Kolmogorov's one. The differences between the frame of KE (1) and Kolmogorov-type turbulence are: a) the presence of a distinguished frequency scale in the system, provided by the peak of spectrum, b) the nonlocality of four-wave nonlinear interactions formatting the entire spectrum, and c) the absence of sources and sinks external to the wave system.



Fourth, the (quasi-) constant fluxes, $P_E(\omega)$ and $P_N(\omega)$, realised in the calculations, and the leakage of total energy $E$ according to ratio (32) or growth of action $N$ to (28) are only the formal mathematical consequences of the accepted versions of the computational algorithm for solving KE (1), rather than the reason of establishing a self-similar spectrum, which in both cases has the same tail (21). To maintain these fluxes, the peak-domain for self-similar spectrum $S_{sf}(\omega,\theta)$ formally plays

the role of internal source or sink, depending on the case considered, whilst the spectrum tail plays the counterpart role.

Fifth, as far as the KE-solution dose not save simultaneously the total energy and total wave action, the problem of choosing the preserved value needs a further research. In our mind, the preservation of total energy $E$ is preferable in the course of KE-solution from the physical point of view, as the system is conservative. This preference is determined by the fact that the conservation of energy is proved for the Euler equations (Zakharov, 1968), whilst the wave action, $N$, is only the

auxiliary variable of the theory, the invariance of which is not prescribed by the Euler equations. From this point of view, the previous results with Kolmogorov-type spectra formation (e.g., Polnikov 1990, 1994; Pushkarev et al. 2003; Badulin et al. 2005), based on solving the extended KE of form (8), require further elaboration by means of additional numerical studying the features of the processes of establishing Kolmogorov-type spectra in forms (6) and (7) both for isotropic and anisotropic initial spectra, sources and sinks.

### Appendix A

Because of the obtaining temporal asymptotes of the peak-frequency for self-similar spectrum, $\omega_p(t)$, is presented in literature in a very concise form (e.g., Pushkarev et al., 2003; Badulin et al., 2005), we represent here the main analytical derivations of the proper ratios of forms (16-17) in more details.

First, consider the situation when the self-similar spectrum is realised under the condition of conservation for the total wave action, $N$. As already noted in the text, in this case, the leak of total energy $E$ takes place. Therefore, we start from the definition for energy change, as a consequence of the nonlinear transfer (NLT) of energy, taking in mid the existence of the selfsimilar spectral shape:

$$dE/dt = \frac{d}{dt}\left(\int S_{sf}(\omega)d\omega\right) = \int (dS_{sf}(\omega)/dt)d\omega = \int Nl(\omega)d\omega \propto (g^{-4}\omega_p^{11}S_p^3)\omega_p. \qquad (A1)$$

Here, in the last proportionality ratio, the self-similarity of NLT $Nl(\omega)$ in form (10) is taken into account, with the NLT-dimension written in parentheses (Polnikov 1989). Now, combining formula (26): $E \sim S_p \cdot \omega_p$, the constancy of $S_p$ (33), and the final expression in the right-hand side of (A1), one obtains the differential equation for $\omega_p(t)$ (which is valid for large time $t$). Indeed, from (26) is follows

$$dE/dt \propto d[S_p(t)\omega_p(t)]/dt = S_p[d\omega_p(t)/dt], \qquad (A2a)$$

On the other hand, Eqs. (33) and (A1) give

$$dE/dt \propto \omega_p^{12}(t)S_p^3 \propto \omega_p^{12}(t) \quad . \qquad (A2b)$$



Finally, combination of (A2a) and (A2b) provide the differential equation for $\omega_p(t)$:

$$[d\omega_p(t)/dt] = \text{const} \cdot \omega_p^{12}(t). \tag{A2c}$$

Eq. (A2c) has the evident solution $\omega_p(t)$ of form (16): $\omega_p(t) \propto t^{-1/11}$.

In the case when $E = \text{const}$, the dependence $\omega_p(t)$ is derived in the similar manner. Now, the equation for the wave-action change with time takes the form

$$dN/dt = \frac{d}{dt}\left(\iint \left[S_{sf}(\omega)/\omega\right]d\omega\right) = \left(\iint [\{dS_{sf}(\omega)/dt\}/\omega]d\omega\right) \propto g^{-4}\omega_p^{11}S_p^3, \tag{A3}$$

where the same dimensional coefficient for $Nl(\omega)$ is used as in (A1), taking in mind the self-similarity of NLT. Furthermore, using the facts that $N \sim (S_p \cdot \omega_p)/\omega_p \sim S_p$, and $S_p \propto \omega_p^{-1}$ (see Eqs. (26-28) in the main text), from (A3) one obtains the proper differential equation for $\omega_p(t)$:

$$dN/dt \propto d[\omega_p^{-1}(t)]/dt \propto \omega_p^{-2}(t)[d\omega_p(t)/dt] \propto \omega_p^{11}(t)S_p^3(t) \propto \omega_p^{8}(t). \tag{A4}$$

From (A4), it follows the solution of form (17): $\omega_p(t) \propto t^{-1/9}$.

It remains to note that the formulas, used to obtain $\omega_p(t)$, in no way depend on the degree of spectrum anisotropy. They are based only on the following assumptions: a) the self-similarity of both the spectrum, $S_{sf}(\omega,t)$, and NLT, $Nl_{sf}(\omega,t)$; b) the conditions for the conservation of total wave action $N$ or wave energy $E$ (i.e., ratios (31) and (26), respectively); and c) the well-known dimension for the NLT.

**Team list**

1. Vladislav Polnikov, Research professor, leading scientist of the A.M. Obukhov Institute of Atmospheric Physics of RAS, Moscow, 119017, Russia, E-mail: polnikov@mail.ru.

Fangli Qiao, Professor, Deputy Director General of the First Institute of Oceanography of SOA, Qingdao, 266061, China, E-mail: qiaofl@fio.org.cn.

Yong Teng, scientist of the First Institute of Oceanography of SOA, Qingdao, 266061, China, E-mail: engyong2003@fio.org.cn

**Author contribution**

V. Polnikov proposed the idea, method and tools of investigation, and executed the analysis and prepared a part of the text.

F. Qiao helped with the results analysis and preparation of the text.

Y. Teng executed computations and made the plots.



## 2. Competing interests

As far as this work competes with the previous results by the Prof. Zakharov's group (e.g., Badulin S, Pushkarev A, Geodjaev V,, Korotkevich A.) and numerous his coauthors (e.g, Resio D, L'vov, Fal'kovich and others, in references), this could make a conflict of interests in the publication process. The same can be said for Van Vledder G, and Perrie W, who use the competing WRT-method for calculating KI.

**Acknowledgements**. The first author is grateful to participants of the session on nonlinear phenomena, have been held at the Shirshov Institute of Oceanology of RAS (December of 2017, Moscow), for their valuable criticisms, which have helped to elaborate the text of this paper. This study was carried out with the partial support of RFBR project No. 18-05-00161. The research was jointly supported by the NSFC-Shandong Joint Fund for Marine Science Research Centers under Grant U1606405.



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
