# Peer review of "Asymptotes of the nonlinear transfer and wave spectrum in the frame of the kinetic equation solution"

_Nonlinear Processes in Geophysics, 2018_

## Short Comment (SC1) · 30 Aug 2018

**Comments on the paper**
**"Asymptotes of the nonlinear transfer and wave spectrum in the frame of the kinetic equation solution" by Polnikov, Vladislav; Qiao, Fangli; Teng, Yong**

The previous version of the paper has been reviewed for *Journal of Fluid Mechanics* 3 months ago. The titles of these two versions differ by the only word (swell -> wave). The present review largely follows my report for the *JFM*.
.
   The authors results are based on numerical solutions of the Hasselmann equation (HE) for the case of spatially homogeneous random wave field with different initial conditions. Authors are making an attempt to relate features of the initial conditions with asymptotic behavior of the collision integral (nonlinear transfer term NLT in words of authors). The goal of searching for such link and the results themselves are clear from the paper abstract and text.

   Authors used the algorithm by Polnikov. This algorithm has never been properly described, tested and compared with alternative numerical approaches. Numerous inconsistencies in the problem statement do not allow qualifying the results as confident. Quality of Excel-made graphics and the corresponding figure captions are unconvincing. The quality of presentation is poor: grammar, improper use of terms, numerous typos and corruption of names.

   In addition to the previous report to *JFM* one can provide more examples of poor quality of the paper:

**In Abstract**

**L18** "asymptote of nonlinear energy transfer becomes negative". "Negative asymptote" sounds, at least, as a slang;

**P2L3 –** $(k_x,k_y)=(k,\theta)$   meaningless (again, a mathematical slang?);

**P2L9 –** Authors used kernels (elements in words of authors) derived by Crawford et al. (1980). As shown by Krasitskii (1994) these kernels are not correct (discussion in page 2 of Krasitskii), they make the Zakharov equations non-Hamiltonian and, hence, do not lead to conservation laws for wave action, energy (formal) and momentum (formal). Thus, further authors' speculations on integrals of motion for eq.1 make no sense;

**P3L19-20 –** "Finally, only two methods have survived …". It is quite questionable declaration. The method of V.I. Lavrenov is always in use (see works of M. Benoit). The new method of Geogjaev & Zakharov (2017) has been recently presented (authors corrupt the first author name throughout the text);

**P3L27 –** "Polnikov (1989)  have showed". The algorithm of Polnikov is incorrect (see above) and "the conservation balance for the integral values" has not been properly investigated in the cited paper;

**P3L32 –** The self-similarity of solutions for the kinetic equation has not been demonstrated by Polnikov (1990, see this paper) because of rather short duration of

simulations. Approximation of the spectral peak frequency decay in this paper by the power law $t^{-0.1}$ (eq.9) contradicts to the analytical result $t^{-1/11}$ (e.g. Baduli et al. 2005);

**P4L25** – Reference Badulin & Zakharov (2017) is absent in the reference list;

**Eq12a** – The limit of wave length in simulations about 3 cm (51 rad/s) looks irrelevant (see below the review for JFM).

The list of notes can be far continued.

In the cover letter authors claim:
**Competing interests**
*"As far as this work competes with the previous results by the Prof. Zakharov's group (e.g., Badulin S, Pushkarev A, Geodjaev V,, Korotkevich A.) and numerous his coauthors (e.g, Resio D, L'vov, Fal'kovich and others, in references), this could make a conflict of interests in the publication process. The same can be said for Van Vledder G, and Perrie W, who use the competing WRT-5 method for calculating KI."*

In the reviewer opinion, the declaration contradicts the journal *Competing interests policy* and looks as an attempt to avoid professional discussion of the paper by excluding the majority of experts in wind-wave studies.

The paper is below any professional standards and cannot be published in *Nonlinear Processes in Geophysics*.

**Report for JFM, May 4, 2018**

**Comments on the paper**
**"Asymptotes of the nonlinear transfer and the swell spectrum in the frame of the kinetic equation"**
**by Polnikov, Vladislav; Qiao, Fangli; Teng, Yong**

The paper is below any professional standards and cannot be published in *Journal of Fluid Mechanics*. The radical opinion of the reviewer can be summarized in three "i"s illustrated by examples. More comments are given in the attached file.

i)   **Ignorance.** Example: Lines 8-10, page 6. *"We used two versions of the numerical algorithm, corresponding to the exact conservation of either wave energy E or wave-action N".*
Conservation laws are inherent properties of the equation under study and cannot be regarded or disregarded by *"versions of the numerical algorithm"*;

ii) **Incompetence.** Example: Eq. 7a, page 5. Frequency ω=80 rad/s corresponds to waves shorter than 2 cm, i.e. to capillary waves that cannot be related to the problem of sea swell (see the manuscript title);

iii) **Irreverence.** The paper is full of disrespectful comments to results of other authors, improper citations. Some authors' names in the list of references are corrupted (e.g. Garganier instead of Gagnaire, Geojaev instead of Geogjaev). Authors do not cite the recent paper "Ocean swell within the kinetic equation for water waves" by Sergei I. Badulin and Vladimir E. Zakharov Nonlin. Processes Geophys., 24, 237–253, 2017  https://doi.org/10.5194/npg-24-237-2017 (please, compare with the manuscript title). They mention other papers of Prof. Zakharov's group as "*a large series of papers by Zakharov and co-authors*" without explicit citations.

The paper cannot be published in *Journal of Fluid Mechanics.*

---

## Author Comment (AC1) · 3 Sep 2018

The letter of reply on the comments by Dr. S. Badulin on manuscript "Asymptotes of the nonlinear transfer and wave spectrum in the frame of the kinetic equation solution" by Vladislav G. Polnikov, Fangli Qiao and Yong Teng

First of all, we share all comments by Dr. Badulin into three types: 1) Editorial, 2) Methodical, 3) Physical. Below, consider them step by step. 1) The first kind of comments (in the notations by Badulin: P3L19-20, about name of Geogjaev, and P4L25, about missing of the reference) are accepted. We thank Dr. Badulin for these comments. Though, all editorial comments do not influence on the results understanding,

and could be easily improved whilst the final edition of the text.

2) The second kind of comments (all the others: about the KI kernel, verification of algorithm, and so on) are the statements not furnished with proofs. (See our supplement also). All the features of the used numerical methods were described and verified by Dr. V. Polnikov (1989, 1990) long before the appearing any calculations by the Zakharov's group (including Dr. Badulin). If researches read the proper papers attentively, they could find the answers to their questions. Indeed, in paper Polnikov (1989) the total checking of the method for the KI estimations was executed, which was never repeated by other authors, including Masuda(1980), Resio and Perrie (1991), Van Vledder (2006), and the Zakharov's group. We have never seen detailed description of their results about accuracy of the KI estimations and the total energy and wave action conservation! The same we can say about details of the algorithm for the KE solution (compare descriptions in Polnikov 1990; Resio and Perrie , 1991; Komatsu and Masuda, 1996).

Thus, we cannot discuss the points of the methods and accuracy remotely. It needs to gather together, moreover, as some authors are living in Moscow, and clarify all details. Face to face. Otherwise, all the statements are not furnished with proofs. Dr. V. Polnikov have addressed to S. Badulin with this proposal several times, unfortunately, without positive reply. It seems that in this point there is a conflict of interests (see the proper remark in our supplement).

In turn, we have several doubts about features of the WRT method used by Badulin (and the Zkharov's group). But it is not a point for the remote discussion. We simply trust them and accept their results for further references (see the list of references). Hereby, we could close this methodical point and address directly to the physical results.

3) Unfortunately, any statements about our physical results and conclusions are simply absent in Dr. Badulin's comments. He said that "Numerous inconsistencies . . .. do
not allow quantifying the results as confident". It means that Dr. Badulin simply has refused of "the discussion in essence". No one of our physical results and conclusion statements was not argued (in no way). As far as the conclusions of the paper are not argued, in fact, we have nothing to discuss.

If somebody repeat our calculations, and obtain the same or other results, it will be a good reason for the discussions. Hope that publication of our paper will stimulate such researches.

In addition to the said, some detailed replies to Dr. Badulin's comments are pasted directly into the text of his comments, and attached as the supplement to this letter of reply.

We hope that all our arguments (said above) will be taken into attention during the professional reviewing the text of our manuscript.

On behalf of the co-authors, Dr. Vladislav Polnikov 03.09.2018

Please also note the supplement to this comment:
https://www.nonlin-processes-geophys-discuss.net/npg-2018-35/npg-2018-35-AC1-supplement.pdf
* * *
[Figure]

**Supplement:**

**Comments on the paper**
**"Asymptotes of the nonlinear transfer and wave spectrum in the frame of the kinetic equation solution"**
**by Polnikov, Vladislav; Qiao, Fangli; Teng, Yong**

The previous version of the paper has been reviewed for *Journal of Fluid Mechanics* 3 months ago. The titles of these two versions differ by the only word (swell -> wave). The present review largely follows my report for the *JFM*.
.

 The authors results are based on numerical solutions of the Hasselmann equation (HE) for the case of spatially homogeneous random wave field with different initial conditions. Authors are making an attempt to relate features of the initial conditions with asymptotic behavior of the collision integral (nonlinear transfer term NLT in words of authors). The goal of searching for such link and the results themselves are clear from the paper abstract and text.

 Authors used the algorithm by Polnikov. This algorithm has never been properly described, tested and compared with alternative numerical approaches. Numerous inconsistencies in the problem statement do not allow qualifying the results as confident. Quality of Excel-made graphics and the corresponding figure captions are unconvincing. The quality of presentation is poor: grammar, improper use of terms, numerous typos and corruption of names.

 In addition to the previous report to *JFM* one can provide more examples of poor quality of the paper:

**In Abstract**

**L18** "asymptote of nonlinear energy transfer becomes negative". "Negative asymptote" sounds, at least, as a slang;

**P2L3 –** $(k_x,k_y)=(k,\theta)$  meaningless (again, a mathematical slang?);

**P2L9 –** Authors used kernels (elements in words of authors) derived by Crawford et al. (1980). As shown by Krasitskii (1994) these kernels are not correct (discussion in page 2 of Krasitskii), they make the Zakharov equations non-Hamiltonian and, hence, do not lead to conservation laws for wave action, energy (formal) and momentum (formal). Thus, further authors' speculations on integrals of motion for eq.1 make no sense;

**P3L19-20** – "Finally, only two methods have survived …". It is quite questionable declaration. The method of V.I. Lavrenov is always in use (see works of M. Benoit). The new method of Geogjaev & Zakharov (2017) has been recently presented (authors corrupt the first author name throughout the text);

**P3L27 –** "Polnikov (1989)  have showed". The algorithm of Polnikov is incorrect (see above) and "the conservation balance for the integral values" has not been properly investigated in the cited paper;

**P3L32 –** The self-similarity of solutions for the kinetic equation has not been demonstrated by Polnikov (1990, see this paper) because of rather short duration of

simulations. Approximation of the spectral peak frequency decay in this paper by the power law $t^{-0.1}$ (eq.9) contradicts to the analytical result $t^{-1/11}$ (e.g. Badulin et al. 2005);
[Figure]

**P4L25 –** Reference Badulin & Zakharov (2017) is absent in the reference list;

**Eq12a –** The limit of wave length in simulations about 3 cm (51 rad/s) looks irrelevant (see below the review for JFM).

The list of notes can be far continued.

In the cover letter authors claim:

**Competing interests**

*"As far as this work competes with the previous results by the Prof. Zakharov's group (e.g., Badulin S, Pushkarev A, Geodjaev V,, Korotkevich A.) and numerous his coauthors (e.g, Resio D, L'vov, Fal'kovich and others, in references), this could make a conflict of interests in the publication process. The same can be said for Van Vledder G, and Perrie W, who use the competing WRT-5 method for calculating KI."*

In the reviewer opinion, the declaration contradicts the journal *Competing interests policy* and looks as an attempt to avoid professional discussion of the paper by excluding the majority of experts in wind-wave studies.

The paper is below any professional standards and cannot be published in *Nonlinear Processes in Geophysics*.

**Report for JFM, May 4, 2018**

***Comments on the paper***
***"Asymptotes of the nonlinear transfer and the swell spectrum in the frame of the kinetic equation"***
***by Polnikov, Vladislav; Qiao, Fangli; Teng, Yong***

The paper is below any professional standards and cannot be published in *Journal of Fluid Mechanics*. The radical opinion of the reviewer can be summarized in three "i"s illustrated by examples. More comments are given in the attached file.

i)    **Ignorance.** Example: Lines 8-10, page 6. *"We used two versions of the numerical algorithm, corresponding to the exact conservation of either wave energy E or wave-action N".*
Conservation laws are inherent properties of the equation under study and cannot be regarded or disregarded by *"versions of the numerical algorithm";*

ii)  **Incompetence.** Example: Eq. 7a, page 5. Frequency $\omega=80$ rad/s corresponds to waves shorter than 2 cm, i.e. to capillary waves that cannot be related to the problem of sea swell (see the manuscript title);

iii)  **Irreverence.** The paper is full of disrespectful comments to results of other authors, improper citations. Some authors' names in the list of references are corrupted (e.g. Garganier instead of Gagnaire, Geojaev instead of Geogjaev). Authors do not cite the recent paper "Ocean swell within the kinetic equation for water waves" by Sergei I. Badulin and Vladimir E. Zakharov Nonlin. Processes Geophys., 24, 237–253, 2017  https://doi.org/10.5194/npg-24-237-2017 (please, compare with the manuscript title). They mention other papers of Prof. Zakharov's group as "*a large series of papers by Zakharov and co-authors*" without explicit citations.

The paper cannot be published in *Journal of Fluid Mechanics.*

---

## Short Comment (SC2) · 9 Sep 2018

Round 2

The reviewer is thankful to the authors for prompt reply. Unfortunately, this reply does not contain answers to specific questions that impedes fruitful discussion. In the round 2 report I would like to focus on just two specific points of the previous review.

P2L9. The authors' use of the Crawford et al. (1980) non-symmetric kernels makes the corresponding version of the kinetic equation (KE) non-conservative. All the conservation laws (energy, action, momentum) cannot be derived from this authors' version

of the kinetic equation!!! Hence, asymptotic solutions (24, 25) cannot be obtained. Hence, further discussion of "Correspondence between numerical and analytical results" (sect. 3.3) makes no sense. The reviewer considers this remark as a solid proof to reject the paper.

The paper Fig.2b reproduced in this report shows a 'power-law fit' of authors' results. This fit is unconvincing in reviewer's opinion because it depends essentially on choice of low- and high-frequency cutoffs. The dependence in log-log coordinates is far from a linear fit (see figure below), i.e. it is far from a power-law approximation in authentic linear axes. The authors have no right to discuss power-law dependencies in this case. This is an additional proof of low quality of the study and a solid reason to reject the paper.

The graph shows $Nl(\omega)$, $m^2$ on the vertical axis (from $1.E-05$ to above $1.E-04$) versus $\omega$, rad/s on the horizontal axis (from 10 to 100), with curves labelled 1 and 1'.

(b)

**Fig. 1.**

---

## Author Comment (AC2) · 11 Sep 2018

The letter of reply on the comments by Dr. S. Badulin "round 2" on manuscript "Asymptotes of the nonlinear transfer and wave spectrum in the frame of the kinetic equation solution" by Vladislav G. Polnikov, Fangli Qiao and Yong Teng

We are thankful to Dr. Badulin for his interest to our paper. But it seems that he did not read the manuscript to the end. It is seen from his comments which do not touch the main results and conclusions of the paper. By the way, the same remark we have made in our first letter of reply to his first letter of comments.

In the second round of comments, Dr. Badulin is continuing to state (P2L9) that using "the Crawford et al. (1980) non-symmetric kernels makes the corresponding version of the kinetic equation (KE) non-conservative". This is the statement which is absolutely not furnished with proofs.

Indeed, in (Polnikov 1989) we have compared our calculations with ones by Masuda(1980) and found very good correspondence. Moreover, in the discussed text it is said that for fast falling spectra (with $n \geq 5$), the conservation of the total wave energy is fulfilled very well. Thus, this point is closed.

For fast falling spectra, the power-falling tail for NL-transfer is well expressed (see Fig. 1b). Herewith, on page 10, lines 11-15, we state that (citation from our text) "5) When spectrum-decay parameter n approaches to 4, the relative intensity of the NLT-tail decreases radically (Fig. 2a); and when $n \leq 4$, the decay-features of NLT begin to depend significantly on the spectrum parameters: $\gamma$, a, relative frequency $\omega/\omega$p (Fig. 2b), and on the limits of the computing grid in units of $\omega/\omega$p . This feature is due to the slow convergence of the KI in such a case (see other numerical details in Polnikov and Uma, 2014)." Just this case was taken by Dr. Badulin to criticize our work. It is the wrong choice.

Dr. Badulin simply did not take into account that Fig. 2b is the explicit demonstration of the radical transition of the form for function NL($\omega$), when parameter n approaches to 4. This is why the power-like tail is not expressed at the first step of the KE solution.

Note again, at the long-term stage of evolutions, the power-like tail of NL($\omega$) is well displayed and has form NI($\omega$)$\sim - \omega$^(-4.15 $\pm$ 0.05) (Eq. 22). For a more cogency, we put the additional Fig. (1+) in the supplement to this reply. The figures of such a kind were not inserted into the text due to their triviality. The proper results for parameter p are presented in Table 2, with the hope that a reader will trust to the authors.

Evidently, that Dr. Badulin has done the improper choice for making his decision about our work (in both comments). It needs to go on and discuss, for example, Table 2, Figs.

6, and 7 (first of all), among others results.

Thus, we are urged to repeat that "the discussion in essence" is still ahead, when somebody will repeat our calculations and obtain the same or other results.

Hope that publication of our paper will stimulate such research.

On behalf of the co- authors, Dr. Vladislav Polnikov. 11.09.2018
* * *
[Figure]

(a)

[Figure]

(b)

Fig. 1+. Asymptote of NL(ω) at the long-term stage of evolution ($t>10^6$ s) .
Bold line is NL(ω); thin line is asymptote.
(a) Run 2 from Table 2 (equation : $y = 6E\text{-}09x^{-4.187}$); (b) Run 8 from Table 2 ($y = 4E\text{-}09x^{-4.139}$)

**Fig. 1.**

---

## Referee Comment (RC1) · Anonymous Referee #1 · 17 Sep 2018

Referee's report on the manuscript "Asymptotes of the nonlinear transfer and wave spectrum in the frame of the kinetic equation solution" by V.G. Polnikov, F. Qiao and Y. Teng

The authors perform numerical simulations of the Hasselmann kinetic equation for various initial spectra without forcing or dissipation. For initial conditions, they take modified JONSWAP spectra with various spectral slopes and directional distributions. It appears that the authors are mostly interested in the slopes (which they call "asymptotes") of spectra and of their time derivatives (the nonlinear transfer function), both for large times and for small times. Considering the nonlinear transfer function at small

times (at the first step of the algorithm), the authors conclude that it depends on the initial conditions. Based on this fairly trivial and non-surprising observation, the authors introduce the concept of "non-locality of nonlinear interactions", which they present as the fundamental property of wave fields, attempting to use it for the interpretation of the subsequent results for large times.

Long-term simulations give mostly familiar results, which add little to the well-established picture of wave field evolution in the framework of the kinetic equation. The quality of the numerics is questionable. While the well-known large time asymptote for peak frequency is one of the most robust features of the Hasselmann equation, the authors are able to find it only rather approximately (Eq. 34). Some of the spectra have rather strange shapes, e.g. in figure 6a for large time. From Table 2, it appears that the directional distribution of an initially isotropic spectrum always remains isotropic (this is also stated in the text, page 15 bottom). Again, the authors draw their fundamental conclusion of "non-locality of the four-wave nonlinear interactions" from this fact. Recently Badulin and Zakharov (Ocean swell within the kinetic equation for water waves, Nonlin. Processes Geophys. 24, 237–253, 2017), having simulated the long-term evolution of nearly isotropic spectra, showed just the opposite, but their paper is not referenced.

Most conclusions of the paper are based on the fundamentally flawed understanding of the kinetic equation properties, and the mix up of physical and numerical realities. The authors use two different versions of the same algorithm, with the imposed conservation of either energy, or wave action. Even from the purely numerical viewpoint, this is rather questionable: a good algorithm should respect the conservation properties of the physical problem under consideration without additional machinations. From the theoretical viewpoint the Hasselmann equation without forcing and dissipation is well known to conserve wave action, but not energy, which slowly leaks to small scales (Badulin et al 2005, referenced in the text). Therefore, a good Hasselmann equation algorithm should conserve wave action with sufficient accuracy; if it doesn't, it is not good enough. The authors prescribe the conservation of wave action artificially, at

each step of the algorithm (page 11). As said above, I do not think this is a good idea; but in another version of the algorithm, they force the conservation of energy, and draw various conclusions from the comparison of the two versions of the algorithm. They claim that in the latter case they observe the flux of wave action towards large scales (inverse cascade). The authors appear to be completely unperturbed by the fact that they present two different numerical models of the same physical reality, with different conservation properties, as equally acceptable. In fact, the inverse cascade as a physical phenomenon is simply not present in their problem, since it would require forcing in high frequencies, which is not present. Actually, by imposing energy conservation, the authors effectively pump some energy into the system at each timestep, therefore replacing the original problem with a different one. Apparently, this "forcing" is too weak to form the "-11/3" spectral slope typical of the inverse cascade.

Sections 4 "Interpretation of results" and 5 "Conclusions", based on the comparison between the two numerical models, contain statements that are either (a) not new (b) incorrect (c) senseless. "The presence of a distinguished frequency scale in the system" (which contradicts the concept of self-similarity, which the authors use extensively themselves) and "total action is only an auxiliary analytical variable" belong to the second category, while "non-locality of four-wave nonlinear interactions", "the spectral peak... plays the proper role of the internal sink or source in this system", "fluxes... are not the reasons for self-similar spectrum formation", "we do not deal with the Kolmogorov-like turbulence", and many others, belong to the third one.

This paper cannot be considered for publication.

---

## Author Comment (AC3) · 18 Sep 2018

The letter of reply on the comments by anonymous reviewer #1 on manuscript "Asymptotes of the nonlinear transfer and wave spectrum in the frame of the kinetic equation solution" by Vladislav G. Polnikov, Fangli Qiao and Yong Teng

This review can be sheared in two parts.

In the first part (the first half of review), the reviewer has made remarks about several details of the paper. But nearly all of them do not hold water. Indeed, 1) In his remark about the non-locality, the reviewer did not notice that we have used the spectra with

the same power-falling tail but different peak parameters. Herewith, the power-falling law of the NL-transfer tail is different. This is not a trivial result. This is not a simple influence of the initial condition. This is the non-locality firstly described here. It seems that the reviewer has not noticed this idea.

2) The remark about "large time asymptote for peak frequency " was commented in the text of our paper. The small difference of powers in Eqs. (29) and (34) is acceptable for numerical calculations, as it is well covered by the statistical scattering for these powers.

3) The remark about Table 2 is simply the draw-away words. We did not state the non-locality from Table 2 results. It was done rather earlier in the text.

4) The only remark about the missing reference (Badilin and Zakharov, 2017) in the list of references is fully accepted. Though, this is the negligible misprint with could be easily improved.

Thus, we may conclude that this part of remarks is not principal for a judging the text merits.

The second part of remarks is more principal one. Here the reviewer has tried to argue the authors approach for this study.

5) The reviewer did state "From the theoretical viewpoint the Hasselmann equation without forcing and dissipation is well known to conserve wave action, but not energy, which slowly leaks to small scales (Badulin et al 2005, referenced in the text)". Yes, we know this paper, which is one among others by the Zkahrov's group, where this result was stated. In our text we fixed this results as the "paradox fact", which takes place in the conservative system (the conservatively of the potential-wave system was proved by Zakharov, 1968). From paper by Zakharov (2017) we know that this leakage of energy ( in the solution of KE ) is due to low convergence of the kinetic integral for slow-falling spectra appearing in this solution. This fact is also mentioned in our paper

as the defect of the present approximation for KE. To overcome this defect of the KE, we accepted two versions of the KE-solutions: with preserved wave action, and with preserved wave energy.

6) The reviewer said "I do not think this is a good idea". In opposite: this is a very good idea. It permits us to check the point of applicability of the Kolmogorov-turbulence treating the results of the KE solutions. This idea has resulted in the unexpected and amazing result. Indeed, in the case of preservation of total wave energy, the long-term asymptote of the KE-solution has the same falling law as in the case of preserving wave action. In this regard, the reviewer makes a comment: "Actually, by imposing energy conservation, the authors effectively pump some energy into the system at each time step, therefore replacing the original problem with a different one. Apparently, this "forcing" is too weak to form the "-11/3" spectral slope typical of the inverse cascade". By these words, in fact, the reviewer has agreed that in the solution of the pure KE, the spectrum with the tail-faling power "-11/3" is not established; despite of the fact that the downward wave-action flux takes place. Thus, the Kolmogorov treating has no place. That is what we want! The reviewer does support our main result.

7) Herewith, the reviewer has finalized his comments by the listing of our conclusions, stating that they are "(a) not new (b) incorrect (c) senseless". These reviewer's statements are not furnished with proof, because the listing of our conclusions, following after these statements, is not a proof. In opposite, these conclusions are new, correct, and have a very important physical sense.

As there are no other principal remarks, we can state that, taking into account the fact of supporting our results by the reviewer (see the end of remark 6), we have a positive result of this discussion.

Thus, the positive final of our discussion with the reviewer permits us to end our reply and hope on the publishing our paper (despite of the negative reviewer's conclusion, which we consider as the homage to the Zakharov's group results).

Some details of our replies to the certain reviewer's remarks are pasted into the text of Review #1, given as the supplement to this our reply.

On behalf of the co- authors, Dr. Vladislav Polnikov. 18.09.2018

Please also note the supplement to this comment:
https://www.nonlin-processes-geophys-discuss.net/npg-2018-35/npg-2018-35-AC3-supplement.pdf

**Supplement:**

Referee's report on the manuscript "Asymptotes of the nonlinear transfer and wave spectrum in the frame of the kinetic equation solution" by V.G. Polnikov, F. Qiao and Y. Teng

The authors perform numerical simulations of the Hasselmann kinetic equation for various initial spectra without forcing or dissipation. For initial conditions, they take modified JONSWAP spectra with various spectral slopes and directional distributions. It appears that the authors are mostly interested in the slopes (which they call "asymptotes") of spectra and of their time derivatives (the nonlinear transfer function), both for large times and for small times. Considering the nonlinear transfer function at small

times (at the first step of the algorithm), the authors conclude that it depends on the initial conditions. Based on this fairly trivial and non-surprising observation, the authors introduce the concept of "non-locality of nonlinear interactions", which they present as the fundamental property of wave fields, attempting to use it for the interpretation of the subsequent results for large times.

Long-term simulations give mostly familiar results, which add little to the well-established picture of wave field evolution in the framework of the kinetic equation. The quality of the numerics is questionable. While the well-known large time asymptote for peak frequency is one of the most robust features of the Hasselmann equation, the authors are able to find it only rather approximately (Eq. 3).
[Figure]
 Some of the spectra have rather strange shapes, e.g. in figure 6a for large time. From Table 2, it appears that the directional distribution of an initially isotropic spectrum always remains isotropic (this is also stated in the text, page 15 bottom). Again, the authors draw their fundamental conclusion of "non-locality of the four-wave nonlinear interactions" from this fact.
[Figure]
 Recently Badulin and Zakharov (Ocean swell within the kinetic equation for water waves, Nonlin. Processes Geophys. 24, 237–253, 2017), having simulated the long-term evolution of nearly isotropic spectra, showed just the opposite, but their paper is not referenced.

Most conclusions of the paper are based on the fundamentally flawed understanding of the kinetic equation properties, and the mix up of physical and numerical realities. The authors use two different versions of the same algorithm, with the imposed conservation of either energy, or wave action. Even from the purely numerical viewpoint, this is rather questionable: a good algorithm should respect the conservation properties of the physical problem under consideration without additional machinations.
[Figure]
 From the theoretical viewpoint the Hasselmann equation without forcing and dissipation is well known to conserve wave action, but not energy, which slowly leaks to small scales (Badulin et al 2005, referenced in the text). Therefore, a good Hasselmann equation algorithm should conserve wave action with sufficient accuracy; if it doesn't, it is not good enough. The authors prescribe the conservation of wave action artificially, at

each step of the algorithm (page 11). As said above,
[Figure]
 I do not think this is a good idea; but in another version of the algorithm, they force the conservation of energy, and draw various conclusions from the comparison of the two versions of the algorithm. They claim that in the latter case they observe the flux of wave action towards large scales (inverse cascade). The authors appear to be completely unperturbed by the fact that they present two different numerical models of the same physical reality, with different conservation properties, as equally acceptable. In fact, the inverse cascade as a physical phenomenon is simply not present in their problem, since it would require forcing in high frequencies, which is not present.
[Figure]
 Actually, by imposing energy conservation, the authors effectively pump some energy into the system at each timestep, therefore replacing the original problem with a different one. Apparently, this "forcing" is too weak to form the "-11/3" spectral slope typical of the inverse cascade.

Sections 4 "Interpretation of results" and 5 "Conclusions", based on the comparison
 between the two numerical models, contain statements that are either (a) not new (b) incorrect (c) senseless. "The presence of a distinguished frequency scale in the system" (which contradicts the concept of self-similarity, which the authors use extensively themselves) and "total action is only an auxiliary analytical variable" belong to the second category, while "non-locality of four-wave nonlinear interactions", "the spectral peak... plays the proper role of the internal sink or source in this system", "fluxes... are not the reasons for self-similar spectrum formation", "we do not deal with the Kolmogorov-like turbulence", and many others, belong to the third one.

This paper cannot be considered for publication.

---

## Author Comment (AC4) · 8 Oct 2018

We have discussed two short comments by Dr S. Badulin, and one Referee comments (only). All the editorial remarks are accepted and improved in the revised text.

The technical remarks about the kernel of kinetic integral and computing algorithms are out of the remote discussion. We have done enough checking of their applicability. Thus, it is left only to discuss the computational results.

Unfortunately, nobody discussed our conclusions in Sect. 3.1. Nobody discussed our results (21) and (22), figures 6 and 7, and other numerous new results presented in

Sections 3.2 and 3.3. Nobody discussed our interpretation of the results (Sect 4.). There were only some critical statements, but all of them were not furnished with proof.

Herewith, at present, not all the points touched in this paper are clear in the task considered. For example, the question about reasons of the energy leakage during KE numerical solution, in the conservative nonlinear wave system, is still needs its own detailed consideration (as it is mentioned in the text).

It seems that the mentioned comments authors cannot say any convincing statements about the presented results without their own computations. We hope that publication of our paper will stimulate the wave community to this action in nearest future.

On behalf of the co-authors, Vlad Polnikov 08.10.2018